

# Ice nucleation efficiency of AgI: review and new insights

Claudia Marcolli[1,2], Baban Nagare[1], André Welti[1,3], Ulrike Lohmann[1]

[1]Institute for Atmospheric and Climate Science, ETH Zurich, Zurich, Switzerland
[2]Marcolli Chemistry and Physics Consulting GmbH, Zurich, Switzerland
[3]Leibniz Institute for Tropospheric Research (TROPOS), Leipzig, Germany

*Correspondence to*: C. Marcolli (claudia.marcolli@env.ethz.ch)

**Abstract.** AgI is one of the best investigated ice nuclei. It has relevance for the atmosphere since it is used for glaciogenic
cloud seeding. Theoretical and experimental studies over the last sixty years provide a complex picture of silver iodide as ice
nucleating agent with conflicting and inconsistent results. This review compares experimental ice nucleation studies in order
to analyse the factors that influence the ice nucleation ability of AgI. We have performed experiments to compare contact and
immersion freezing by AgI. This is one of three papers that describe and analyse contact and immersion freezing experiments
with AgI. In Nagare et al. (Nagare, B., Marcolli, C., Stetzer, O., and Lohmann, U.: Comparison of measured and calculated
collision efficiencies at low temperatures, Atmos. Chem. Phys., 15, 13759 – 13776, doi:10.5194/acp-15-13759-2015, 2015)
collision efficiencies based on contact freezing experiments with AgI are determined and compared with theoretical
formulations. In a companion paper, contact freezing experiments are compared with immersion freezing experiments
conducted with AgI, kaolinite, and ATD as ice nuclei. The following picture emerges from this analysis: The ice nucleation
ability of AgI seems to be enhanced when the AgI particle is on the surface of a droplet, which is indeed the position that a
particle takes when it can freely move in a droplet. Ice nucleation by particles with surfaces exposed to air, depends on water
adsorption. AgI surfaces seem to be most efficient as ice nuclei when they are exposed to relative humidity at or even above
water saturation. For AgI particles that are totally immersed in water, the freezing temperature increases with increasing AgI
surface area. Higher threshold freezing temperature seem to correlate with improved lattice matches as can be seen for AgI-
AgCl solid solutions and $3AgI \cdot NH_4I \cdot 6H_2O$, which have slightly better lattice matches with ice than AgI and also higher
threshold freezing temperatures. However, the effect of a good lattice match is annihilated when the surfaces have charges.
Also, the ice nucleation ability seems to decrease during dissolution of AgI particles. This introduces an additional history and
time dependence of ice nucleation in cloud chambers with short residence times.

## 1 Introduction

Cloud glaciation is an important process that influences cloud optical properties, cloud lifetime, and precipitation. While cloud
droplets freeze homogeneously below about 237 K, ice nucleating particles (INPs) are needed to freeze droplets at higher
temperatures. Different modes of ice nucleation are discriminated. Probably the most important process is immersion freezing
where an INP immersed in a water or solution droplet initiates freezing. Similar to this mode, condensation freezing denotes





ice nucleation that occurs concurrent with the activation of an INP to a cloud droplet. Originally, contact freezing referred to freezing initiated by the collision of an aerosol particle with a supercooled droplet (Ladino Moreno et al., 2013). The importance of the collision in the process of contact freezing was challenged by Durant and Shaw (2005) who observed that freezing efficiency is also enhanced when the nucleating particle does not collide with the droplet but is partially immersed in it. To discriminate this nucleation process from traditional collisional contact freezing, it is termed contact freezing inside-out (Durant and Shaw, 2005; Shaw et al., 2005; Fornea et al., 2009; Murray et al., 2012; Gurganus et al., 2013; 2014). Immersion, condensation, and contact freezing are considered different mechanisms of ice nucleation because there is some evidence that they occur with different efficiencies. A number of laboratory studies suggests that droplets freeze at higher temperature in contact mode than in immersion mode (Ladino Moreno et al., 2013). However, many of these studies were not well constraint in terms of the number of collisions between particles and droplets (Ladino Moreno et al., 2013). Deposition nucleation is considered to be different from these three processes because it takes place below water saturation when ice nucleates on a substrate from vapor with no liquid water involved. This conception was recently challenged by Marcolli (2014) who hypothesized that at conditions below water saturation, liquid water condenses in pores because of the inverse Kelvin effect (Sjogren et al., 2007) and freezes eventually by immersion nucleation or homogeneously when the temperature is low enough.

When Vonnegut (1947) detected that a piece of silver iodide is able to freeze liquid water below -3.5°C, he attributed this high ice nucleation ability to the close fit between the crystal lattices of ice and AgI. However, the importance of lattice match was questioned thereupon because ice seems to nucleate preferentially at steps on the silver iodide surface and the hexagonal pattern only appears when the growing ice crystal has gained some size (Zettlemoyer et al., 1961). Crystals such as cadmium sulfide (CdS), quartz, indium antimonide (InSb), and barium fluoride (BaF2) are ineffective as ice nuclei (IN) despite similar lattice spacing with ice (Edwards and Evans, 1962; Sadtchenko et al., 2002; Conrad et al., 2005). Moreover, examples of organic ice nuclei like crystalline steroids nucleate ice almost as effectively as AgI but their crystal lattices exhibit no obvious relationship to the cell dimensions of ice (Head, 1961). The importance of lattice match was further questioned when it was detected that the silver iodide surface is mainly hydrophobic with isolated hydrophilic patches. It was then conjectured that indeed these hydrophilic sites, which might be chemical defects, steps in the crystal lattice or hygroscopic contaminants (Corrin and Nelson, 1968; Zettlemoyer et al., 1963) are the locations where ice nucleation occurs rather than perfect faces of AgI crystals. On the other hand, the structures of some of the most efficient IN closely match the ice lattice. Water freezes as high as -1°C when covered by a monolayer of long-chain alcohols forming 2D crystals with close lattice match to ice (Popovitz-Biro et al., 1994; Majewski et al., 1995). Ice active proteins expressed by the bacteria Pseudomonas syringae (P. Syringae) and Erwinia herbicola which are ice active up to -2°C possess sites with close fit to the ice lattice (Kajava and Lindow, 1993; Yankofsky et al., 1981; Govindarajan and Lindow, 1988; Budke and Koop, 2015). Therefore, a good IN–ice lattice match seems to be one of the requirements that promotes ice nucleation, but it is certainly not the only one. Other requirements may be a low surface charge and large polarizability of the ice nucleating surface. Further factors are the orientation of hydrogen bonds and van der Waals interactions between the IN and water molecules (Sadtchenko et al., 2002; Zielke et al., 2015). Fletcher (1959) pointed out that



the molecular alignment of water molecules directed by hydrogen bonds must be favorable for the growth of further ice layers, otherwise, epitaxial ice growth is inhibited.

AgI has become one of the best investigated ice nuclei. It is of relevance for the atmosphere because of its use for glaciogenic seeding of clouds to increase precipitation and to prevent hail (Silverman, 2001). Theoretical and experimental studies over the last sixty years provide a complex picture of silver iodide as ice nucleating agent with conflicting and inconsistent results. We have performed experiments to compare contact and immersion freezing yielding results contrasting with earlier studies. These new experiments are described here and in a companion paper (Nagare et al., 2016). The aim of this review is to compare experimental ice nucleation studies with AgI and to see whether seemingly conflicting results can be brought into agreement when the nucleation conditions are considered in more detail. Such an analysis should also allow to identify factors that influence the ice nucleation efficiency of silver iodide. These findings can help to optimize procedures for cloud modification with AgI as artificial seed. Moreover, this survey may show exemplarily what factors determine the efficiency of IN in general. This is one of three papers that present and analyze contact freezing experiments with AgI. In Nagare et al. (2015) collision efficiencies based on contact freezing experiments with AgI are determined and compared with theoretical formulations. In a companion paper (Nagare et al., 2016), contact freezing experiments are compared with immersion freezing experiments conducted with AgI, kaolinite, and ATD as INP.

## 2 Literature review

Tables 1, 2, and 3 compile setups and procedures of immersion, contact, and condensation freezing experiments performed with silver iodide. The information given by the different studies is very diverse, which often limits direct comparisons. Some studies report the onset of freezing, which is usually significantly higher than a mean or median freezing temperature. The number or concentration of particles is often missing, preventing the evaluation of surface area present in a droplet. Figure 1 shows median freezing temperatures $T_M$ for frozen fractions ranging from $0.2 - 0.8$ as a function of surface area for the studies that provide enough information to determine or to estimate these quantities. The presented data is coded with respect to freezing mode (open symbols: contact freezing; filled symbols: immersion freezing) and studies (different colors and symbols). As a general trend, Figure 1 shows an increase of $T_M$ with increasing silver iodide surface area present per droplet. However, there are also many exceptions, showing that surface area is an important but not the only factor determining nucleation temperature.

### 2.1 Particles and droplets deposited on substrates

The data from Edwards et al. (1962) given as red circles in Fig. 1 show a clear dependence of freezing temperature on surface area. The AgI particles used in this study were prepared by slowly precipitating AgI with seed solutions containing AgI particles of different concentration, so that crystals grew on the seed particles and no nucleation took place. This procedure yielded monodisperse AgI suspensions of single crystals with particle diameters of $0.17 \pm 0.02$ µm, $0.75 \pm 0.1$ µm, and $3.5 \pm$



0.4 µm. The larger particles crystallized as hexagonal pyramids, smaller ones were more rounded. The particles were deposited onto hydrophobic glass cover slips to which they adhered. Droplets of water or aqueous solutions were sprayed on the slips, which were then flooded with paraffin oil to prevent evaporation. The samples were cooled at a rate of 2 K/min and $T_{50}$, the temperature at which 50 % of the droplets with sizes between 30 and 60 µm had frozen, was reported. Generally, all droplets

froze within about 4 K. The droplets contained $10^{-4}$ M $AgNO_3$ to offset the natural tendency of silver iodide prepared by this method to adsorb iodide ions on the surface. The same general procedure was also used by Edwards and Evans (1962; 1968). AgI crystals prepared by Edwards and Evans (1968) had average diameters of 4 µm (2 – 5 µm) and a crystal habit of hexagonal plates. Droplets sprayed on cover slips contained between 1 and 1000 particles. The $T_{50}$ values of this study given as red triangles in Fig. 1 are in fair agreement with the previous study by Edwards et al. (1962). Edwards and Evans (1962)

investigated the dependence of freezing temperature on excess $Ag^+$ and $I^-$ ions, which adsorb to the surface of AgI particles and influence their surface charge. To do this, they diluted a saturated AgI solution once with a 0.2 M KI solution containing the seed aerosol (sol A) and once with a 0.2 M $AgNO_3$ solution (sol B) leading to excess $I^-$ (orange triangles) and excess $Ag^+$ (orange circles), respectively. It can be seen from Table 1 and Figure 1 (symbols overlap), that in KI solutions with concentrations $> 10^{-5}$ M and $AgNO_3$ solutions with concentrations $> 10^{-3}$ M the freezing temperatures drop, manifesting a clear

negative influence of the surface charge on the nucleation ability of silver iodide.

Gokhale and Goold (1968) performed contact nucleation experiments by sprinkling AgI particles on supercooled droplets on a hydrophobic plate. They observed that the particles (5 – 400 µm in diameter) remained on the surface of the drops and initiated freezing at the initial stage temperature of 268 K. However, they did not quantify the number of particles present, which precludes an evaluation in terms of surface area. They performed similar experiments for an AgI smoke produced from

an AgI string generator with particle diameters from 50 – 100 nm. These particles initiated freezing of 50 % of droplets at 263 K when the stage was cooled at a rate of 1.3 K/min. Gokhale and Goold (1968) concluded that these freezing temperatures are 5 – 10 K higher than the ones observed by Hoffer (1961) for droplets embedded in an oil with immersed AgI particles and attributed it to an enhanced freezing probability for dry particles on a surface compared with particles immersed in the droplet. However, a strict comparison is not possible because in both studies, information is lacking to quantify the surface area present

per droplet. In a follow-up study, Gokhale and Lewinter (1971) monitored the freezing process of 2 mm water droplets with a movie camera and observed that nucleation was initiated at the point of particle contact and continued from there over the entire surface of the drop. The interior of the drop froze at a much slower rate.

Zimmermann et al. (2007) studied heterogeneous ice nucleation by AgI in an environmental scanning electron microscope by increasing the water pressure in the sample chamber at constant temperature. First ice formation in the condensation mode was

observed at a temperature of 268 K, and in the deposition mode at 264 K. These threshold temperatures are in good agreement with those of Schaller and Fukuta (1979, see below).





## 2.2 Experiments with droplets embedded in oil

Purple triangles and purple squares in Fig. 1 represent immersion freezing data of droplets in emulsions from Zobrist et al. (2008) and Aguerd et al. (1982), respectively. AgI particles were prepared in situ by precipitating a silver nitrate solution and a potassium iodide solution in the presence of an oil containing a surfactant for emulsification. The freezing experiments were

carried out in a differential scanning calorimeter (DSC) immediately after preparation. The droplet size reported in Zobrist et al. (2008) and Aguerd et al. (1982) was $3 \pm 2$ µm and $< 3$ µm, respectively. In the DSC thermograms of Zobrist et al. (2008) and Aguerd et al. (1982) a heterogeneous and a homogeneous freezing peak appears. The presence of the homogeneous freezing peak could be either due to the absence of crystallized AgI in some of the droplets or inactive AgI crystallites. To quantify the surface area present per droplet, crystallization of one spherical particle per droplet was assumed, yielding particle

diameters of 200 – 500 nm. For Zobrist et al. (2008) we assigned the onset of the heterogeneous freezing peak to freezing of 5 µm droplets and the maximum to freezing of 3 µm droplets. To quantify the surface area of AgI per droplet for Aguerd et al. (1982), an average droplet size of 2 µm was assumed. Considering the uncertainties in estimating surface areas per droplet, the results from emulsion freezing experiments are in good agreements with each other and with the study performed with deposited AgI crystallites from Edwards et al. (1962). Aguerd et al. (1982) also investigated a 1 cm$^3$ bulk sample by mixing

0.01 M AgNO$_3$ and 0.01 M KI, for which they determined freezing at 269.5 K.

Hoffer (1961) investigated freezing of 100 – 120 µm water droplets embedded in a silicone oil. AgI particles were produced from the reaction of silver nitrate with sodium iodide. He observed a median freezing temperature of 257 K when the droplets were cooled at a rate of 1 K/min and concluded that this value is lower by 9 K compared with results from cold chambers. However, with the given information, a strict comparison is not possible because the surface area of AgI present in a droplet

cannot be quantified.

## 2.3 Cloud chamber experiments

Sax and Goldsmith (1972) performed contact and immersion freezing experiments in a cloud chamber. Freely falling droplets with diameters of 40 – 160 µm (average: 100 µm) intercepted a horizontal aerosol stream of $5 \cdot 10^6$ cm$^{-3}$ AgI particles with 30 nm diameter (size range from 10 – 40 nm) for 0.04 s (1 cm in vertical extent). The aerosol was produced by heating an AgI-

coated resistance wire to T = 700°C in a nitrogen stream. For contact freezing experiments the droplets were brought in thermal equilibrium before intercepting the aerosol stream. After coagulation with the AgI particles, the droplets proceeded into an observation chamber where frozen droplets were distinguished visually from liquid ones. Coagulation of 100 µm droplets with 30 nm particles were dominated by Brownian motion. Assuming a collision efficiency of ca. 0.3, around 100 particles would be captured by the droplet (note that this number is higher than the collection of only 1 particle estimated by

Sax and Goldsmith, 1972). For immersion freezing experiments, the droplets passed the aerosol stream at T > 273 K, before they were cooled to the target temperature. Residence time in the chamber was around 4 s. Immersion freezing occurred at 2 K lower temperature than contact freezing.



DeMott (1995) investigated the efficiency of ice formation with silver iodide-type aerosols depending on ice nucleation mode with the aim to reproduce ice seeding conditions as closely as possible. Therefore, he used a field-scale solution combustion device. This generator produces AgI aerosols by burning AgI-acetone-ammonium iodide-water solutions in a propane flame. To produce AgI-AgCl aerosols 20 – 30 mole% ammonium perchlorate has to be added to the standard solution. Contact freezing experiments were performed in an isothermal cloud chamber, which isolates an artificially generated cloud at a supercooled temperature. The cloud is continuously replenished during an experiment to maintain a constant cloud droplet concentration. Total nominal droplet concentrations were either 2100 cm$^{-3}$ or 4300 cm$^{-3}$ in individual experiments. Results are shown in Fig. 1 for 30 nm and 70 nm particles as black open squares. For immersion freezing experiments, clouds were formed on AgI-AgCl aerosols by continuous expansions at T > 268 K, leading to freezing at 257 K, as indicated by black filled squares in Fig. 1. DeMott (1995) concluded that immersion freezing is more than one order of magnitude less efficient than contact freezing for AgI-AgCl aerosols.

Langer et al. (1978) investigated contact freezing of monodisperse AgI aerosols in a cloud chamber. Silver iodide aerosol was generated thermally by passing dry air over molten AgI at temperatures up to 100°C above the AgI melting point of 558°C. A NaCl aerosol was used to generate the liquid cloud in the chamber. During operation, there was a continuous stream of AgI aerosol and NaCl aerosol. The nucleating fraction and freezing temperatures were observed to increase with increasing particle size. Particles with diameters < 20 nm and < 10 nm were essentially inactive at temperatures of 259 K and 253 K, respectively. Schaller and Fukuta (1979) investigated deposition, condensation, and contact freezing of AgI in their ice thermal diffusion chamber. AgI aerosol was produced by heating an AgI powder on an iron plate. The smoke was introduced into the chamber and frozen fractions were evaluated after 1 min nucleation time. They observed deposition nucleation up to 264 K and condensation freezing up to 268.5 K (frozen fraction of 1.3 % after 1 min). The onset temperature and the frozen fraction for condensation freezing increased with increasing supersaturation. Contact freezing was observed from 265 – 267 K. This temperature range nearly coincides with the one for condensation freezing. For comparison, Schaller and Fukuta (1979) performed an experiment where the AgI aerosol was activated above the ice nucleation threshold temperature. When the fog had formed, the chamber was cooled. No difference in ice nucleation behavior was observed for this procedure compared to condensation freezing experiments.

Edwards and Evans (1960) investigated condensation freezing of an AgI aerosol in a cloud chamber. They noted that supersaturation of 20 % with respect to water was needed to activate the AgI particles with typical diameters of 12 – 20 nm. Condensation freezing occurred up to 267 K. The activity of this aerosol was also tested when it was deposited on a substrate. At 98 % RH and temperatures of 262 K, 259 K, and 255 K a small fraction of particles (<0.005) was active in deposition mode.

## 2.4 Bulk experiments

Vonnegut and Baldwin (1984) investigated the kinetics of ice nucleation by measuring the time to freezing at constant temperature of a bulk sample with immersed AgI particles of 1 µm diameter. While freezing occurred with an average





nucleation time of 1 s at 264 K, it took more than 10 000 s to nucleate the sample at 270 K. This shows the stochastic nature of ice nucleation and that nucleation time has to be considered together with freezing temperature and surface area for comparisons between different studies.

Palanisamy et al. (1986b) investigated bulk samples in test tubes cooled at a rate of 0.1 K/min and observed freezing of pure

AgI particles at 270.82 K and of AgI-AgCl cubic solutions even higher at 271.97 K.

Davis et al. (1975) reported freezing of water in contact with $3AgI \cdot NH_4I \cdot 6H_2O$ films at 272 K compared with freezing at 269 K for AgI films.

## 2.5 Continuous flow diffusion chamber experiments

Nagare et al. (2015; 2016) presented contact freezing experiments performed with the ETH CoLlision Ice Nucleation CHamber

(CLINCH) using 80 µm diameter droplets, which were exposed to 200 nm and 400 nm diameter AgI particles. These particles were prepared by mixing $AgNO_3$ and KI solutions. The suspension was atomized and the droplets were dried and size selected. Residence times in the chamber which was held at ice saturation were 2 s or 4 s. For aerosol concentrations of 500 – 2000 cm$^{-3}$ low freezing efficiencies with median freezing temperatures of only 247 – 248 K were measured. At these concentrations, the droplets collected on average less than one particle. For a particle concentration of 5000 cm$^{-3}$ and 4 s residence time, the

droplets collected on average 2.35 particles and the median freezing temperature was 258 K. Immersion freezing experiments in the IMCA/ZINC chamber (see Appendix A1) with an AgI aerosol prepared by the same method as for the CLINCH experiments have also been carried out. In one set of experiments, particle size was varied for droplet residence time of 10 s. For particles with diameters of 40 – 400 nm, median freezing was observed at ca. 264 K almost independent of particle size. For 30 nm particles, the median freezing temperature decreased to 259 K and the heterogeneously frozen fraction leveled off

at ca. 0.9, indicating that 10 % of the particles did not induce ice nucleation. For particle diameters of 20 nm, most droplets froze homogeneously. Langer et al. (1978) showed that 20 nm particles produced by heating AgI induced contact freezing at 253 K. Therefore, the small size cannot be the reason why ice nucleation decreased drastically for particles < 40 nm. Indeed, comparing the size distribution obtained by atomizing the AgI suspension with the one obtained by atomizing pure Milli-Q water (Appendix A2, Fig. A1) reveals that 30 nm diameter particles produced by atomization of the AgI suspension contained

a minor fraction of particles originating from the Milli-Q water and the 20 nm particles are even dominated by these particles. For 200 nm and 400 nm AgI particles, experiments with 3 s residence time were also carried out. This shorter residence time decreased the median freezing temperature by 1 – 2 K.

## 3 Discussion

Figure 1 shows a general trend of increasing freezing temperatures with increasing AgI surface area, however, there are also

several exceptions. For a more consistent comparison, instead of freezing temperatures, nucleation rate coefficients expressed in units of e.g. cm$^{-2}$s$^{-1}$ should be compared. However, most studies do not provide enough information to determine rates.



Freezing temperatures as a function of surface area depend on measurement specific factors like the frozen fraction they refer to, the nucleation mode, and properties of the AgI particles dependent on the preparation method. In the following, relevant factors influencing heterogeneous ice nucleation by AgI will be discussed in more detail.

### 3.1 Freezing temperature and frozen fraction

The freezing temperatures, which should characterize the efficiency of an IN, are not independent of instrumental factors that have to be taken into account when results from different instruments are compared. Some detection methods provide onset temperatures of freezing. These depend on the detection limit of the measurement technique and are higher than median freezing temperatures, which are also often reported. With the IMCA/ZINC setup, freezing temperatures for frozen fractions of 0.1 – 1 are measurable. For 200 nm AgI particles and 3 s residence time, a median freezing temperature of 262 K was

obtained. A frozen fraction of 0.1 was already reached at 265 K and a frozen fraction of 1 only at 254 K. This illustrates the differences that can be expected when freezing temperatures refer to different frozen fractions.

Classical nucleation theory predicts a dependence of nucleation rates (nucleation events per time) on the surface area present in a sample due to the stochastic nature of the nucleation process. Therefore, the freezing temperature is expected to increase with increasing surface area. Even stronger dependence on surface area is expected when active sites are responsible for ice

nucleation because the probability to contain an efficient active site increases with increasing surface area. This dependence is taken into account in Fig. 1 by plotting freezing temperatures as a function of surface area. For immersion freezing, the stochastic nature of nucleation also leads to a dependence of freezing temperatures on time. Vonnegut and Baldwin (1984) showed that the freezing temperature increased from 264 K to 270 K when the observation time was extended from 1 s to 2 h. For contact freezing, the number of collisions during the experiment has to be taken into account to convert frozen fractions

to freezing efficiencies (e.g. Nagare et al., 2016). For this, the collision efficiency has to be known, a quantity that depends on many factors such as size and charge of particles and droplets, temperature and water vapor concentration gradients between droplet surface and the surrounding (e.g. Nagare et al., 2015). For contact freezing inside-out, there is a dependence of freezing temperatures on nucleation time, as for immersion freezing. Since collisional contact freezing is supposed to be immediate after a particle collides with a droplet, it is unclear how to properly compare the nucleation efficiencies of collisional contact

freezing and immersion freezing.

### 3.2 Preparation method of AgI

The preparation method of AgI influences the size, polymorphic form, morphology, and number of surface defects of the produced particles. Most procedures lead to mixtures of the stable β- and the metastable γ-form (see Appendix B1). Commercial AgI is usually the β-form. The basal 001 and 00-1 faces of the β-form show a close lattice match with ice and

are often considered responsible for the ice nucleation ability of AgI. These faces correspond to the -1-1-1 and 111 faces of the γ-form, respectively. Typical crystal habits of the γ-form expose these faces partly to the surface (see Appendix B2).



Therefore, the condition of lattice match is realized also for the γ-form, so that this form can induce ice nucleation as well as the β-form if lattice match is important. This was confirmed in a recent molecular dynamics study by Zielke et al. (2015) who found that β-AgI and γ-AgI were able to nucleate ice but only on the silver exposed surface (see Appendix B4 for more details). Edwards et al. (1962) prepared monodisperse silver iodide particles and noted that particles of the same size are not equally

active as INP, but behave as if the number of nucleation sites per particle is proportional to the surface area of the particle. This led them to the conclusion that ice nucleation does not occur on the whole surface but on surface irregularities, which might be preferred nucleation sites. They suggested that the occurrence of nucleation sites depends only on the surface area irrespective whether there were few large or many small particles in their sample. However, depending on the preparation method, AgI crystals may contain different amounts of surface irregularities. Shevkunov (2005) conjectured that particles

produced by solution combustion and used for cloud seeding have a superior ice nucleation ability compared to crystals with perfect surfaces, because by this method particles contain many surface defects. Nevertheless, from the immersion freezing studies compared in Fig. 1, such a relationship is not evident: freezing temperatures from experiments carried out with single crystals (Edwards et al. 1962) align well with freezing temperatures determined from emulsion experiments (Zobrist et al., 2008; Aguerd et al., 1982) conducted with irregularly shaped particles from in-situ precipitation (see Appendix A2). Note that

surface irregularities might be more important when the particles are exposed to air, if the ice nucleation efficiency depends on the amount of water that adsorbs at surface defects or impurities (see discussion below).

When AgI is prepared by precipitation from $AgNO_3$ and KI solutions, the counterions $K^+$ and $NO_3^-$ remain in solution and form conglomerates with AgI when the suspension is dried, e.g. during atomization (see Fig. A2 in Appendix A). $KNO_3$ dissolves again when the particle is in contact with water. The freezing ability might be altered during dissolution. However,

it is rather unlikely that an effect persists once the soluble salts are completely dissolved. The dilution of dissolved salts in the droplets is large enough so that the freezing point depression by the solute is negligible.

There is a strong influence of freezing temperatures on surface charge, which depends on the stoichiometric ratio of $Ag^+$ to $I^-$ ions. This charge effect has a strong influence on the ice nucleation ability of AgI particles and will be discussed below in Sect. 3.3.

**3.3 Lattice match and surface charge**

The closeness of fit between the ice lattice and the lattice of the ice nucleating substance is often considered as a basic requirement for heterogeneous ice nucleation. Indeed, AgI surfaces show a close lattice match with ice. The fit is even improved for AgI-AgCl cubic solid solutions and the complex compound $3AgI \cdot NH_4I \cdot 6H_2O$ (see Appendix B1). Davis et al. (1975) reported a misfit for the basal plane of β-AgI with respect to the basal plane of ice of 1.5 % at 266 K and an even lower

misfit with respect to the complex compound $3AgI \cdot NH_4I \cdot 6H_2O$ of 1.3 %. This is consistent with the higher threshold freezing temperatures that they observed for $3AgI \cdot NH_4I \cdot 6H_2O$ compared with AgI. Namely, heterogeneous ice nucleation on films of these IN occurred at 272 K for $3AgI \cdot NH_4I \cdot 6H_2O$ compared with 269 K for AgI. Palanisamy et al. (1986b) determined ice



nucleation threshold temperatures of bulk samples of AgI-AgCl solid solutions with different AgI:AgCl mixing ratios and compared them with lattice constants from X-ray powder diffraction. The 65:35 mol% combination of AgI and AgCl showed the best lattice match of 0.44 % and the highest nucleation temperature of 271.97 K in bulk experiments. Experiments with AgI, yielded a freezing temperature of 270.62 K and a lattice match of 1.77 %. AgCl has a large misfit of 12.86 %, nevertheless, nucleation was observed at a threshold temperature of 269.13 K (Palanisamy et al., 1986b). A superior ice nucleation ability of AgI-AgCl INP compared with AgI particles has also been observed by DeMott (1995) for experiments in the CSU isothermal cloud chamber.

While these studies support the importance of lattice match for the threshold freezing temperature, in other studies, the necessity and importance of lattice match is questioned. Conrad et al. (2005) explored homogeneous ice nucleation and found that the critical embryo that caused ice to form bears little structural similarities to hexagonal ice, Ih. Therefore, he suggested that the site for heterogeneous ice nucleation does not have to be ordered or more particularly, does not have to possess a hexagonal geometry to match the structure of Ih. In a molecular dynamics study, Fitzner et al. (2015) have found that lattice match is at most desirable but not a requirement for heterogeneous ice nucleation.

If there is an effect of a good lattice match, it can be annihilated by a high surface charge. A study by Edwards and Evans (1962) showed freezing temperatures that were highest for zero net charge and clearly lower in the presence of either excess $Ag^+$ or $I^-$ ions in the solution. Fletcher (1959) considered the importance of entropy for the growing ice embryo: a positive or a negative surface charge leads to a reduction of freezing efficiencies because the charge orders OH dipoles of water molecules parallel to one another and therefore reduces the entropy and raises the free energy of the growing ice embryo. The basal planes of AgI either end with $Ag^+$ or with $I^-$ ions. If ice grows on a basal face, the oxygens will be above ions which all have the same sign. Therefore, for lowest interfacial energy, the OH dipoles of the ice molecules at the interface must all be parallel leading to low configurational entropy and in sum to a high free energy of nucleation. The prism face exposes equal numbers of $Ag^+$ and $I^-$ on the surface. Therefore, there is no entropy effect on the prism face and Fletcher (1959) concluded that this face of silver iodide should be superior at nucleating ice compared with the basal face, although its lattice match is inferior with a misfit of 1.6 % compared to the value of 1.4 % for the basal face (Fletcher, 1959). This conclusion was objected by Fukuta and Paik (1973), who calculated potentials of water molecules adsorbed on AgI surfaces which indicated that the entropy effect of ice nucleation can be ignored on AgI surfaces because the maximum configurational entropy can be easily realized by the forming ice germ. In agreement with this, simulations by Fraux and Doye (2014) and Zielke et al. (2015) were able to simulate ice growth at the silver terminated basal face of β-AgI but neither on the iodide terminated basal face nor on the prism face (see Appendix B5 for more details). The superiority of the $Ag^+$-terminated compared with the $I^-$-terminated basal face was ascribed to the chair conformation of water molecules on the $Ag^+$-terminated face, which resembles the chair conformation present in hexagonal and cubic ice (Zielke et al., 2015). Both studies found that ice did not grow directly on the substrate but that a templating mechanism may be active so that particular AgI surfaces impose a structure in the adjacent water layer that closely resembles a layer that exists in bulk ice (hexagonal or cubic). Ice nucleates on this layer and not directly on the AgI surface. Similarly, Taylor and Hale (1993) suggested that ice nucleation occurs within a thicker film of adsorbed water on the





basal AgI face after the two-dimensional nucleation of a solid-like monolayer adjacent to the substrate. The three dimensional nucleation would then occur on the ice-like underlayer with an extremely small lattice mismatch. However, such a templating mechanism does not rule out that defects such as steps and ledges or impurities are also important (Zielke et al., 2015; Taylor and Hale, 1993).

## 3.4 Position of particles in or on the droplet

Durant and Shaw (2005) and Shaw et al. (2005) showed that freezing temperatures for ice nucleating particles that contact the water surface are higher than those of totally immersed particles. Depending on the experimental setup, particles can be forced in a position either in or on the droplet or they can be free to take the energetically most favorable position. In the studies performed by Edwards et al. (1962) and Edwards and Evans (1962; 1968) particles stick to the substrate and are totally immersed in the droplets. In the emulsion freezing studies by Zobrist et al. (2008) and Aguerd et al. (1982) droplets are within the oil phase so that the particles within the droplets have no access to the air-water interface.

If particles are floating in a droplet or bulk volume with free access to the air-water interface, the position of the particles depends on their wetting behavior with water. In the absence of other forces, they become totally immersed when they are completely wetted by water (contact angle of zero), otherwise they float on the surface (Nagare et al., 2016). For AgI the (intermediate advancing) contact angle is 45° – 50° (see Appendix B5) indicating that particles adhere to the water surface. This is in accordance with our own observations that AgI particles remain on the surface when gently sprinkled on water and with the observations by Gokhale and Goold (1968) and Gokhale and Lewinter (1971) when they sprinkled AgI particles on drops.

When an ice nucleating particle is activated to a cloud droplet, it is usually considered to be totally immersed in the droplet. However, when an AgI particle is free to take the energetically most favorable position, it will adhere to the surface of a droplet. Analysis of immersion freezing studies in Fig. 1 suggests that freezing temperatures are higher for those studies, where the particle is free to take a position on the surface of the droplet as is the case for experiments performed with IMCA/ZINC or with a cloud chamber as in the study by Sax and Goldsmith (1972).

During contact freezing experiments carried out in the CLINCH chamber, particles should remain on the surface of the droplets after they collided with a droplet. For immersion freezing experiments in IMCA/ZINC and contact freezing experiments in CLINCH, the same AgI preparation procedure was used, the same particle sizes were selected and residence times in the chambers were comparable. Therefore, similar freezing temperatures could be expected in both experiments. However, median freezing temperatures were around 16 K lower for contact freezing experiments in the CLINCH chamber than for immersion freezing experiments in the IMCA/ZINC chamber. Possible explanations for this will be discussed in the sections below.

## 3.5 Water adsorption and surface irregularities

The reason why freezing temperatures are increased when the IN is positioned at the air-water interface compared with totally immersed is still debated. Gokhale and Lewinter (1971) conjectured that particles on the surface are better at nucleating ice





than totally immersed particles because of two likely reasons: first, water molecules probably have less freedom of movement on the surface than molecules within the drop since they are more or less bound into a polarized layer at the surface. An oriented layer of this nature should be more susceptible to nucleation than disoriented bulk water. Second, the dissipation of latent heat is much easier at the surface than within the bulk water. Another explanation, that has gained increasing attention

recently, is that the energy barrier is reduced at the three-phase contact line of water, air, and nucleating surface (Djikaev and Ruckenstein, 2008; Gurganus et al., 2014).

How much water is adsorbed on the AgI surface depends on the relative humidity and the presence of surface irregularities. The water uptake of a defect-free AgI surface was calculated to be below one nominal monolayer at water saturation. Even above water saturation, water gathers in patches and does not form a monolayer also at nominal monolayer coverage (see

Appendix B4). More water can adsorb on a surface with irregularities such as defects and impurities, compared with a perfect surface. In this case, water patches form in and around these irregularities and might be the starting point for ice nucleation.

If ice nucleation occurred indeed on such water patches on the AgI surface, ice nucleation efficiency should depend on RH. Contact freezing experiments in CLINCH were performed at ice saturation and immersion freezing experiments in IMCA/ZINC at water saturation. Therefore more water is adsorbed on the AgI surfaces in IMCA/ZINC experiments what

indeed might increase the freezing temperature (for water adsorption on AgI surfaces see Appendices B4). Condensation freezing studies showed that the highest freezing temperatures are obtained for high supersaturation during activation (Schaller and Fukuta, 1979; Zimmermann et al., 2007), which may increase the water present at the surface of the particle. There is evidence that ice crystals form on specific sites (Bryant et al., 1959). These sites might be surface irregularities like dislocations and steps where water uptake is enhanced. Bryant et al. (1959) observed that at T = 261 – 269 K ice crystals formed at selected

sites when the air was supersaturated with respect to water. At lower temperatures it was sufficient when the air was supersaturated with respect to ice. Bassett et al. (1970) conjectured that ice nucleation is proceeded by water adsorption at discrete locations which might be polar impurities on a hydrophobic surface leading to cluster formation and rapid migration of weakly adsorbed molecules to the growing clusters. Freezing of some of the clusters should occur before the surface of the substrate is totally covered by a liquid-like layer of water. When the substrate also exhibits good fit with ice, as in the case of

silver iodide, the interfacial energy is reduced and the nucleating ability should be maximized. Thus a disordered cluster of adsorbed water molecules may not offer as much resistance to freezing as bulk water. Once formed, growth of the ice island is facilitated by rapid diffusion of molecules weakly adsorbed on the rest of the surface (Bassett et al., 1970).

### 3.6 Solubility and dissolution of AgI

Silver iodide is a sparingly soluble salt with a solubility product of around $10^{-16}$ at 298 K (Hiemstra, 2012). The solubility

decreases with decreasing temperature and reaches a value $< 10^{-17}$ at 273 K (Lyklema, 1966). Nevertheless, a small particle in a large droplet will dissolve to some degree. In the CLINCH chamber (Nagare et al., 2015; 2016), a 200 nm particle will dissolve by around 2.4 % when it collides with an 80 µm droplet. In the IMCA/ZINC setup, particles with diameters between 20 – 200 nm were activated to droplets of 20 µm, leading to partial dissolution of 0.04 % for a 200 nm particle and 38 % for a



20 nm particle. Effective dissolution might even be larger, because the particles prepared by precipitation also contain shares of $KNO_3$ (see Appendix A2).

Partial dissolution of a particle changes its shape and surface. Steps, corners, and kinks of atomic dimensions are eroded away and pits below a critical size form and disappear spontaneously. If they have reached the critical size, they continue to grow (Conrad et al., 2005). Pits are preferentially generated along or in dislocation boundaries. Calculations by Conrad et al. (2005) demonstrated that pitting improved the ice nucleation ability of a $BaF_2$ surface.

During the dissolution process, nucleation may be impeded because the eroding surface is changing too fast to let an ice embryo grow. Moreover, dissolution may lead to (additional) surface charge (Waychunas, 2014). A transient charge during dissolution may result when $Ag^+$ and $I^-$ ions dissolve at different rates. Klimeš et al. (2013) have recently shown a preference for dissolution of $Cl^-$ ions over $Na^+$ ions for dissolving NaCl crystals. The $Cl^-$ ions are released first as this exposes more $Na^+$ ions at the surface creating favorable adsorption sites for water. A similar effect may also influence the charge on dissolving AgI surfaces because of the better solubility of $Ag^+$ compared to that of $I^-$ (Hiemstra, 2012). However, precise predictions are not possible because aqueous dissolution processes are not well understood on the molecular scale yet (Waychunas, 2014).

Time scale of dissolution can be calculated with the Noyes-Whitney equation (see Appendix B6). Using this equation, particles that collide with droplets in CLINCH will keep dissolving during the residence time of 2 s and 4 s in the chamber. On the other hand, droplets activated in the IMCA/ZINC instrument probably reach saturation while still in the IMCA section so that there is no ongoing dissolution in the ZINC chamber where ice nucleation should occur. If dissolution is indeed impeded on dissolving crystals, this difference might explain the superior ice nucleation ability of AgI particles in IMCA/ZINC compared with CLINCH. If in contact freezing experiments, more than one particle collides with a droplet, dissolution will be strongest for the first particle and less for subsequent ones. Indeed, for the highest particle concentration of 5000 $cm^{-3}$ in CLINCH, more than two particles collided on average with a droplet. This may explain the higher freezing temperature for this concentration compared with lower ones.

If ice nucleation on dissolving AgI particles is reduced, the experiments performed by Sax and Goldsmith (1972) should also be affected by dissolution because in these experiments the residence time in the chamber was on the same timescale as dissolution (about 4 s).

## 3.7 Weather modification by cloud seeding

The era of weather modification started in the 1940s with first field experiments with the intention to stimulate rain from convective clouds by glaciogenic seeding (Silverman, 2001). Since then many operational programs and scientific studies have been conducted for rain enhancement and hail suppression (Chen and Xiao, 2010; Breed et al., 2014). In these programs silver iodide smokes are produced and injected into clouds mostly with ground based methods, but also with airplanes or rockets. Although in many countries weather modification programs are conducted, considerable skepticism exists as to whether these methods indeed provide a cost-effective means for increasing precipitation for water resources (Bruintjes, 1999). The ability to influence and modify cloud microstructure in certain simple cloud systems such as fog and simple orographic clouds has



been demonstrated in observational studies (Bruintjes, 1999). Nevertheless, the general impact of glaciogenic seeding is still uncertain since it is a challenge to detect a signal from seeding in the noisy pattern of precipitation (Pokharel et al., 2014). Cloud seeding strategies have been established for seeding of cold clouds. There is a need to adapt and optimize seeding strategies according to the different situations. For this, it is required to improve the understanding of the cloud seeding

procedure based on laboratory findings and numerical models (Garstang et al., 2005).

Most AgI aerosols used operationally for weather modification are produced by solution combustion or pyrotechnic generation methods (DeMott, 1995). Atomized AgI-acetone solutions are burned in a propane flame. During subsequent cooling in the atmosphere aerosol particles form. Usually, instead of pure AgI, a mixture of AgI and AgCl is used to produce AgI-AgCl solid solutions with improved ice nucleation ability compared with AgI. Large laboratory cloud chambers have been used to

determine the ice nucleation effectiveness of aerosols for weather modifications (DeMott et al., 1983). The optimal sublimation temperature for silver iodide was found to be 750°C. The optimal suspension concentration in acetone was determined to be 2 wt % (Shevkunov, 2005).

Combustion produces particles with diameters of around 30 nm with only few particles reaching sizes of 100 nm. It is unlikely that these small particles act as cloud condensation nuclei (CCN), because the Kelvin effect opposes the activation to droplets.

To enhance the water uptake of silver iodide particles, NaCl is often added to the acetone solution to increase the hygroscopicity of the particles and to improve their ability to act as CCN. When such particles freeze during CCN activation, this is considered as ice nucleation in condensation mode. If clouds are seeded from an airplane, supersaturation with respect to water is too low for CCN activation and the silver iodide particles need to collide with cloud droplets to act as ice nucleating particle. Collisions of such small particles with cloud droplets occur mainly by Brownian diffusion (Seinfeld and Pandis, 2006). If the droplets

bear high charges, the attraction by electrostatic forces may increase the number of collision between silver iodide particles and cloud droplets. The analysis presented in this study suggests that the ice nucleation ability of dissolving AgI particles is reduced. Moreover, the freezing temperature increases with increasing surface area. Therefore, high enough concentrations are needed to saturate cloud droplets with AgI for glaciogenic seeding in the case of low supercooling.

**Conclusions**

Ice nucleation by silver iodide has been the subject of many experimental and theoretical studies. Although it is an artificial ice nucleus, it has relevance for the atmosphere since it is used for glaciogenic cloud seeding. In this review, physical properties of AgI have been analyzed with special attention to the ones that might be important for heterogeneous ice nucleation. Freezing temperatures as a function of surface area of AgI have been evaluated. The following factors seem to be relevant for heterogeneous ice nucleation on AgI particles:

-    The ice nucleation ability of AgI seems to be enhanced when the AgI particle is on the surface of a droplet compared with totally immersed. This is indeed the position that an AgI particle takes when it can freely move in a droplet.





- For AgI particles partly exposed to air, the ice nucleation ability seems to be influenced by surface water. Water adsorption increases with increasing relative humidity and is enhanced in the presence of surface defects. Indeed, some of the highest freezing temperatures have been observed for condensation freezing when cloud droplet activation occurred at high supersaturation with respect to water.

- For AgI particles that are totally immersed in water, the freezing temperature increases with increasing AgI surface area. There is no evident influence of number of surface defects. Surface defects might be more important when water adsorption on particles exposed to air is involved in ice nucleation.

- Higher threshold freezing temperatures seem to correlate with improved lattice match. AgI-AgCl solid solutions and $3AgI \cdot NH_4I \cdot 6H_2O$ have slightly better lattice matches with ice than AgI and their threshold freezing temperatures are
also slightly higher.

- Edwards and Evans (1962) showed a clear dependence of ice nucleation ability on surface charge with highest freezing temperatures close to the point of zero charge and lower ones for positively and negatively charged surfaces.

- No influence of the polymorphic form on the ice nucleation ability of AgI could be detected. Most preparation methods yield mixtures of the stable β- and the metastable γ-form with different shares. No correlation becomes
evident when the prevailing form expected for a given preparation procedure is compared with freezing temperature. This is in accordance with the results of recent modeling studies that surfaces of the β- and the γ-form are able to nucleate ice (Zielke et al., 2015).

- The ice nucleation ability might be decreased when AgI particles are dissolving. AgI is a sparingly soluble salt. If only one particle is present in a large cloud droplet, dissolution is nevertheless relevant. During dissolution, particles
might acquire surface charge which reduces the ice nucleation ability. Dissolution of nanoscale droplets should occur on the time scale of seconds to minutes. When saturation solubility is reached dissolution levels off and nanoscale pits acquired during dissolution might be preferred nucleation sites. This introduces an additional history and time dependence of ice nucleation in cloud chambers with short residence times.

**Appendix A**

**A1 IMCA/ZINC experiments with AgI**

A suspension of AgI particles was prepared by mixing 0.1 M KI and $AgNO_3$ solutions in Milli-Q water (18.2 MΩ). Particles were generated by atomizing the produced suspension and subsequently dried (as described in Nagare et al., 2015). X-ray diffractograms showed that a mixture of the β- and γ-forms precipitated, which needed days to transform into the β-form. The size distribution of the particles peaks at diameters of about 80 nm as shown in Fig. A1. At particle diameters < 100 nm the
concentration of particles present in the Milli-Q water increases and becomes dominant for diameters < 30 nm. Figure A2





shows STEM (scanning transmission electron microscopy) images of representative AgI particles. The precipitated particles are of irregular shapes and consist of conglomerates of AgI and KNO$_3$.

Immersion freezing experiments were carried out with the IMCA/ZINC setup, which combines the cloud droplet activation chamber IMCA with the ice nucleation chamber ZINC (Lüönd et al., 2010). The IMCA chamber is kept at T > 273 K and RH up to 120 % for activation of all particles. The activated droplets grow to 18 – 20 μm diameter within about 15 s when they pass through the IMCA chamber. The droplets cool down while they pass over to the ZINC chamber, where they reach the target temperature. The residence time in the ZINC chamber, which is kept at water saturation, can be varied between 1 – 21 s by changing the position of the IODE (Ice Optical Depolarization) detector (Nicolet et al., 2010) and the additional sheath air in the ZINC chamber (see Welti et al., 2012). Panel (a) of Fig. A3 shows frozen fractions for particles with diameters between 20 – 400 nm and residence times of 10 s, panel (b) shows a comparison of 3 s and 10 s residence times for 200 nm and 400 nm AgI particles.

## A2 Characterization of AgI particles prepared by precipitation

In the CLINCH and IMCA/ZINC studies, aerosol particles were produced by atomizing a suspension of AgI particles prepared by precipitation from KI and AgNO$_3$ solutions. To decrease the concentration of dissolved KNO$_3$, the supernatant solution was removed and replaced by water as described in Nagare et al. (2015). The suspensions were used for ice nucleation experiments for the next few days. X-ray diffraction of the dried precipitate revealed that it consists of a mixture of the β-phase and γ-phase in similar shares that transformed within less than a week completely into the β-phase when kept in suspension. A TSI 3081 scanning mobility particle sizer (SMPS) was used to measure the particle size distribution, which could be fitted with a lognormal distribution with median diameter of about 80 nm. Transmission electron microscopy (TEM) and high angle annular dark field (*HAADF*) scanning transmission electron microscopy (*STEM*) showed a predominance of particles in the 100 – 500 nm diameter size range, but additionally also the presence of particles with diameters down to 20 nm. The electron microscopy images together with energy dispersive X-ray spectroscopy (EDX) of selected particles shown in Fig. A2 reveal that the particles are conglomerates of AgI and KNO$_3$ crystallites. The primary particles exhibit sizes with diameters ranging from about 20 – 300 nm. When conglomerated particles become immersed in a droplet, KNO$_3$ will dissolve and the primary particles may be released or a conglomerate of AgI may persist.

## Appendix B

## B1 Polymorphic forms

Silver iodide exists in three polymorphic forms. The α-modification represents a body-centered cubic (bcc) lattice. It is the stable form at temperatures above 147°C and melts when heated to 555°C (Burley, 1963). The hexagonal β-AgI is of wurtzite-type and shows a hexagonal closest packing of iodine atoms (Cava and Rietman, 1984). It is thermodynamically stable under



ambient pressure and temperature and transforms to the high temperature stable α-modification at 147°C. The γ-modification represents a face-centered cubic lattice (fcc) with zinc-blende type crystal structure. It is presumably metastable at all temperatures but can be preserved as powder at room temperature for days. It transforms to the hexagonal form within hours when the temperature is kept between 100°C and 147°C and to the high temperature stable α-form at T > 147°C (Berry, 1967).

In stoichiometric ratio, silver iodide $Ag^+$ and $I^-$ ions have both four nearest neighbors. The bonding is partly covalent. The structure can be described as two interpenetrating hexagonally close-packed sublattices, to either give the characteristic [AB] stacking sequence of the hexagonal β-AgI or the cubic stacking sequence [ABC] of the zinc blende structured γ-AgI (Morgan and Madden, 2011). Both the hexagonal and low-cubic phases show a closest packing of iodine atoms with tetrahedral and octahedral holes. Half of the tetrahedral sites but no octahedral sites are occupied by silver. Whether β-AgI or γ-AgI is realized

can be influenced by other cations present as impurities or small shares. Excess cations will enter available interstitial sites, either tetrahedral or octahedral, depending on the ionic radius (Burley, 1963). An excess of cations in the tetrahedral sites should stabilize the hexagonal phase, while an excess in the octahedral sites should stabilize the low-cubic phase. Cations with radii between 0.53 and 0.97 Å such as $Fe^{2+}$, $Li^+$, $Cu^{2+}$, $Na^+$ are able to occupy tetrahedral sites; cations with radii between 0.97 and 1.71 Å such as $K^+$, $Ba^{2+}$, $NH_4^+$, $Cs^+$ may occupy octahedral sites (Burley, 1963).

Co-precipitation of AgI with other silver halides yields solid solutions in which part of the iodide is substituted by the other halide ion. Co-precipitation with AgBr and AgCl leads to a contraction of the AgI lattice due to the smaller ionic radii of $Br^-$ and $Cl^-$ ions compared with $I^-$ (Vonnegut and Chessin, 1971; Palanisamy et al., 1986a, 1986b). X-ray powder diffraction of AgI-AgCl solid solutions have shown that even small impurities of AgCl lead to cubic solid solutions (Palanisamy et al., 1986b). In the presence of ammonium, the complex compound $3AgI \cdot NH_4I \cdot 6H_2O$ may form which has a better epitaxial fit with respect to ice than has silver iodide. The unit cell of this phase is monoclinic and its stability range includes the temperature

range 253 – 282 K at water saturation (Davis et al., 1975).

The commercial powder (Sigma-Aldrich) is of the hexagonal β-form (Guo et al., 2006). By applying pressure by grinding, the powder transforms partly to the γ-modification (Burley, 1963). Precipitation of $AgNO_3$ and KI in stoichiometric ratio at room temperature usually leads to mixtures of the β- and γ- modification (Burley, 1963; Berry, 1967). Manson (1955) e.g. reports

36 ± 8 % β-phase in the precipitate. The only conditions of precipitation which produced entirely hexagonal crystals were when gelatin was present and the solutions were added slowly without an excess of either $Ag^+$ or $I^-$ (Berry, 1967). Faster precipitations give mixtures of the cubic and hexagonal phases, and crystals whose average diameter was as small as 15 nm (Berry, 1967). β-AgI is preferentially produced in the presence of an excess of iodide ions. In the presence of excess silver ions γ-AgI is the predominant form (Sidebottom et al., 1976; Burley, 1963). If the precipitate is kept in solution, the γ-

modification transforms to the stable β-form. Quenching AgI that has been heated to 550°C in air or water produces primarily the metastable γ-form, while slow cooling results in stable β-AgI (Burley, 1963). When AgI is formed by condensation of silver and iodine vapors from the gas phase, 73 ± 2 % β-phase forms when the vapors are heated to 650°C and even 95 ± 2 % β-phase is obtained when the vapors are heated to 800°C or higher (Manson, 1955). This can be explained by excess iodine



present during crystallization: Since silver has a very low vapor pressure even at l000°C, it will condense within the source tube to a large extent, whereas iodine, which boils at 183°C, will be carried out in the effluent and be present in excess. Silver iodide with high shares of γ-AgI can also be obtained by lengthy grinding in a mortar and pestle or by the application of hydrostatic pressure. Complete conversion to the low-cubic form appears to be impossible (Burley, 1963). Tomaev et al. (2012)

reports that the mechanical modification of β-AgI nanocrystals of 500–1000 nm in size stimulates the faceting of the crystals without a significant change in their size and the formation of smaller β-AgI crystals with a characteristic size of 40 nm on their surface. Precipitation of AgI from KI and $AgNO_3$ with the experimental procedure of Nagare et al. (2015) led to a mixture of β-AgI and γ-AgI that also contained $KNO_3$ (see Appendix A2). The γ-modification transformed to the β-modification within days as was determined by XRD.

**B2 Crystal faces and morphology**

A cut along the basal plane of β-AgI creates the 00-1 face with triply coordinated $I^-$ ions and the corresponding 001 face with triply coordinated $Ag^+$ ions. Following the definitions of Hiemstra (2012), the 111 face of γ-AgI terminates with $I^-$ ions and has the same structure as the 00-1 face of β-AgI. The -1-1-1 face terminates with $Ag^+$ ions and has the same structure as the 001 face of β-AgI. The fully loaded 111/-1-1-1 face of γ-AgI can perfectly match the fully loaded 001/00-1 face of β–AgI.

Both faces have the same site density and basic structure, enabling twinning of the two crystal types. Neutral surfaces can be created by the removal of triply coordinated ions from the lattice. With the sites and site densities of these faces, it is not possible to describe the asymmetric charging of AgI crystals, suggesting that they do not contribute significantly to the variable charge behavior of silver iodide. When β-AgI is fractured in parallel to the lateral faces of the periodicity cell exposing the 100 face, ions of both signs are present on the fracture surface. In contrast to the fracture along the bases, there will be no features

of hexagonal symmetry on the fracture surface and the electric field of ions is less uniform (Shevkunov, 2005). The surface of the 100 face of β-AgI is neutral because $Ag^+$ and $I^-$ are present in equal amounts but it can attain a positive charge by adsorbing $Ag^+$ ions, which will protrude from the surface (Hiemstra, 2012). The same is valid for the 110 face of γ-AgI because it is structurally the same. Applying surface complexation modeling to the 100 face, Hiemstra (2012) came to the conclusion that negative charge is created by desorption of $Ag^+$ from the surface leading to positive charge above the surface and negative

charge on the surface, thus, a charge separation is introduced.

Single crystals of β-AgI form hexagonal plates or prisms with 100 faces at the edge (lateral face) and 001/00-1 faces on the planar side (basal face) (Hiemstra, 2012). Depending on growth conditions, they can also form triangular plates (Burley, 1963). In the presence of hexagonal steps hexagonal pyramids may form presumably by a screw mechanism (Burley, 1963). Crystals of γ-AgI may form tetrahedrons terminated by the 111 face or dodecahedrons with dominant 110 faces beside a contribution

of 111 faces (Hiemstra, 2012). Octahedra indicate γ-AgI (Sidebottom et al., 1976). Moreover, the twinning of β-AgI and γ-AgI is possible because of the same surface structure of the planar (00-1/001) face of β-AgI and the 111/-1-1-1 face of γ-AgI as can be observed for large crystals (Hiemstra, 2012). The dominant 110 face of γ-AgI has the same type of structure as the





edge (100) face of β-AgI (Hiemstra, 2012). When γ-AgI particles approach spherical morphology, the 110 face and its equivalents are dominant and the 111-type faces are minor. In the case of equal distances for the opposing crystal faces, the calculated contribution of the 110-type faces is 74%. For hexagonal crystals of β-AgI, the contribution of the dominant edge faces is 67% at equal distances for all opposing surfaces (Hiemstra, 2012).

## B3 Surface charge in water and aqueous solutions

Because of the better solubility of $Ag^+$ compared to $I^-$, a neutral AgI crystal will be negatively charged in water or aqueous solutions. The negative particle charge can be diminished by the addition of extra $Ag^+$ ions, which may readsorb depending on the solution concentration. Adsorption of $Ag^+$ ions leading to positive charge above the surface and negative charge on the surface, induces an electrical double layer consisting of an inner and an outer (second) Stern layer. At a certain $Ag^+$ concentration, equal numbers of $Ag^+$ and $I^-$ are present at the particle surface. Under this solution condition, the particle is in its point of zero excess adsorption or point of zero charge (PZC) (Hiemstra, 2012). The PZC describes the charge of the inner layer (first Stern layer). The charge of the outer layer (second Stern layer) is in addition influenced by the presence of $OH^-$ and $H_3O^+$ ions and depends therefore on pH. The origin of the charge of interfacial water is usually attributed to the higher affinity of $OH^-$ ions, with respect to $H^+$ ions, for the accumulation at the interface (Kallay et al., 2012). At high pH the diffuse double layer is negatively charged and at low pH it is positively charged. Zero charge is reached at the isoelectric point (IEP). The PZC depends only on the concentrations of $Ag^+$ or $I^-$ ions (pAg or pI with pAg + pI yielding the solubility product of AgI) because the $Ag^+$ and $I^-$ ions are charge-determining, whereas the proton is indifferent. Lyklema and Golub (2007) determined the PZC by the titration of an AgI suspension which they had kept for two months in a solution with pI = 4. They obtained PZC = 5.8 (instead of 5.6 which is mostly cited in literature) and did not find a dependence on solution pH. Kallay et al. (2012) measured the IEP, which is also influenced by the charge present due to absorbed $OH^-$ and $H_3O^+$ ions in the second Stern layer. They precipitated AgI particles by gradually adding a KI solution to a silver nitrate solution until the equivalence point was reached and the particles settled. The precipitated particles were washed and dried. They determined the IEP by adding the AgI powder in portions to a dilute potassium iodide solution at pH 6 yielding 4.1 for $KNO_3$ concentrations of $10^{-2} – 10^{-3}$ mol/l and at pH 3 giving 4.6 ($10^{-2}$ mol/l $KNO_3$) and 4.8 ($10^{-3}$ mol/l $KNO_3$). These results support the hypothesis that in neutral environment around pH = 7, the negative charge of water layers at silver iodide surfaces significantly contributes to the electrokinetic charge of the particles, but does not significantly affect the (inner) surface potential. Hiemstra (2012) applied advanced surface complexation modeling (SCM) to the AgI surface and concluded that the inner Stern layer has a very low capacitance in accordance with strong orientation of primary hydration water in the local electrostatic field of the $Ag^+$ and $I^-$ ions. Due to the high water ordering at the AgI surface, the secondary hydration water in the outer Stern layer is also rather strongly structured (Hiemstra, 2012). It is more difficult to charge the AgI surface positively by $Ag^+$ adsorption than to render it negative by uptake of $I^-$ ions (Bijsterbosch and Lyklema, 1978). The $Ag^+$ activity at the PZC always exceeds the



corresponding I$^-$ concentration by many decades. Ag$^+$ ions show a stronger tendency to bind NO$_3^-$ ions than adsorbed I$^-$ ions to bind cations.

## B4 Water adsorption and ice nucleation

In the sixties, adsorption isotherms of water on AgI were measured by several groups and compared with the absorption of inert gases (N$_2$ Kr and Ar) by applying the BET relation. Studies were performed for different preparation methods, degree of sample purity and at several temperatures. Tcheurekdjian et al. (1964) measured adsorption isotherms at T = 259 – 293 K of AgI samples obtained by sublimation, precipitation, for a sample treated with ammonia, and also a commercial sample. They observed that water coverage is only ¼ of the argon area at the nominal monolayer value. Based on this, they concluded that the surface of AgI is largely hydrophobic possessing isolated hydrophilic sites. The entropy of the adsorbed molecules is high, indicating considerable lateral movement. The measurements performed by Hall and Tompkins (1962) at T = 228 – 248 K confirmed the hydrophobic nature of the AgI surface. Corrin and Nelson (1968) investigated water adsorption for T = 257 – 270 K of AgI prepared by silver and iodine reacted in vacuo and purified in liquid ammonia solution. Applying the evaluation procedure suggested in Tcheurekdjian et al. (1964) to this samples resulted in a hydrophilic character of 6 – 9 %. The maximum in isosteric heats was reached for coverages below one monolayer, which was interpreted as a "pseudomonolayer" effect corresponding to cooperative adsorption in patches. Corrin et al. (1964) investigated water adsorption at 303 K on an AgI sample prepared by the reaction between metallic silver and iodine and subsequent treatment with liquid ammonia and a material prepared by precipitation. They concluded that it is quite unlikely that a statistical monolayer of water exists on the surface of AgI at relative pressures less than unity. Higher values are probably due to small amounts of hygroscopic impurities on the AgI surface. Barchet and Corrin (1972) measured adsorption isotherms of a fine powder of AgI that should be free of hygroscopic ionic impurities but with particles containing numerous surface dislocations. The shape of the isosteric heat curves indicated that the adsorbed water was distributed in patches on the surface. Water vapor adsorption isotherms at -3.0°C showed a rapid increase in adsorption when the water vapor pressure raised above ice saturation but remained finite even at water saturation. The isotherms recorded at -6.5 and -10.0°C were terminated by nucleation of the ice phase. Because the surface excess at nucleation clearly increased with temperature, Barchet and Corrin (1972) concluded that the number of embryos which formed on the water adsorbing sites as well as the number of sites did not increase with temperature but that the embryos themselves were larger at higher temperatures.

Barnes (1963) analyzed the NMR (nuclear magnetic resonance) signal of the protons of water molecules on an AgI powder saturated with water vapor to investigate water adsorption and ice nucleation. He found no hysteresis between increasing and decreasing temperature and therefore suggested that all the water present is adsorbed on the surface and not as capillary condensate in pores or pore-like spaces. Moreover, all the adsorbed water layers seemed to be able to freeze at the same temperatures. Barnes and Sänger (1961) investigated by proton spin resonance spectroscopy that the adsorbed water on silver iodide transforms in a temperature range of 20 K from liquid-like to solid-like with an onset of ordering close to the threshold temperature of heterogeneous ice nucleation on silver iodide.



Taylor and Hale (1993) simulated two water layers adsorbed on a model silver iodide (iodine exposed) basal face from 150 to 425 K using Monte Carlo methods. They observed a complex temperature dependence of the structure of the two-layer water condensate where quasi-crystalline, quasi-liquid, and liquid states coexisted at the same temperature. Both layers appeared to be solid at the lowest temperature studied. For T > 265 K the upper layer became increasingly liquid-like with increasing

temperature, whereas the lower layer of water molecules remained generally solid-like up to T = 325 K. The lower layer consisted of five- and six-membered rings centered on the iodine ring with dipole moments preferentially oriented parallel to the substrate and nearly zero dipole moment projection perpendicular to the substrate. This indicates that at least in the second adsorbed layer ice formation above about 265 K is inhibited by the free surface entropy. For the AgI prism face, the first water monolayer is even less structured than on the basal AgI face. Possible remedies to this difficult situation for ice formation on

AgI are not lacking. Some preliminary simulations for two water layers on a model basal face ledge indicate that the ledge is marginally better at promoting ice structure in the upper layer.

Shevkunov (2005) simulated the nucleation of water vapors on the 001 face of β-AgI (basal face) by the Monte Carlo method in a humid atmosphere at 273 K. They found that already below the liquid water saturation pressure, the crystal surface is covered with a water monolayer. The molecules in this layer are arranged above the iodine ions, between the silver ions, and

are interconnected with hydrogen bonds forming the regular structure with hexagonal symmetry. Deposition of a second water layer needed supersaturation, indicating that the condensation of water vapor into a macroscopic phase is prevented by a high free energy barrier. A point defect in the form of an extra ion on the surface did virtually not distort the hexagonal structure of the first water monolayer.

In a follow-up study, Shevkunov (2008) performed Monte Carlo simulations of water vapor nucleation on the basal plane

(001) surface of a β-AgI crystal with and without defects. They concluded that the matching minimizes disorder in layer structure and therefore enhances stability of the layers, promoting layer-by-layer growth. The most stable orientations of molecules in monolayers adjacent to the substrate are parallel and perpendicular to it. The formation of each layer involved overcoming a nucleation barrier. The film thickness is an increasing function of pressure. As the saturation pressure is exceeded, the vapor explosively condensed into a dense macroscopic phase via layer-by-layer growth. The barriers that

suppress the nucleation of successive monolayers can be eliminated by substrate surface defects. However, a point defect did not significantly change the macroscopic behavior of the condensed phase. Extended defects, such as cracks or fractures, should be much more effective in this respect.

Zielke et al. (2015) used molecular dynamics simulations at a supercooling of 20 K to investigate ice nucleation on surfaces representing faces of β-AgI and γ-AgI. These simulations showed that a good lattice match with ice is insufficient to predict

the ice nucleation ability of an AgI surface. When the basal planes of hexagonal ice are compared, only the 001 face terminated by $Ag^+$ ions nucleates ice and not the iodide terminated 00-1 face, although both faces have identical lattice matches with ice. This was ascribed to the configuration that the surface imposes on the water molecules. On the silver exposed surface, the oxygen atoms of the water molecules take a chair conformation, which resembles the chair conformation present in hexagonal and cubic ice. In contrast, for the iodide exposed surface the hexagonal rings form a more coplanar structure that does not



match any surface of ice, and no ice nucleation occurred on this face. Moreover, cubic and hexagonal ice was observed to nucleate on both, the β-form and the γ-form of AgI. On the silver and iodide terminated prism face, no ice could be observed. The authors speculate that a templating mechanism may be active in cases of close lattice match so that particular AgI surfaces impose a structure in the adjacent water layer that closely resembles a layer that exists in bulk ice (hexagonal or cubic). Ice

nucleates at these surfaces and grows almost layer-by-layer into the bulk. They emphasize that this mechanism does not rule out defects being important for ice nucleation by silver iodide, but indicates that defects are not necessary in order to explain why silver iodide particles can be highly effective ice nuclei.

Performing simulations at supercooling of 10 K, Fraux and Doye (2014) also came to the conclusion that ice nucleates on the silver terminated basal surfaces of β-AgI but not on the iodide terminated ones. The ice formation occurred via a two-step

mechanism. A hexagonal ice-like bilayer of strongly adsorbed water molecules immediately formed on the surface. After a certain lag time, a nucleation event occurred and ice started to grow on the top of this layer. Although a somewhat similar adsorbed layer formed on the I⁻-terminated surface, this layer was unable to initiate ice growth. Similarly, although water molecules strongly adsorbed on the prism faces, these layers are less obviously ice-like and were unable to act as templates for ice growth on the time scales of the simulations.

**B5 Contact angle of water on silver iodide**

The contact angle of polycrystalline AgI depends on the silver concentration of the wetting liquid. Billett et al. (1976) measured the contact angle of an air bubble against thin films of silver iodide in an electrolyte solution using the captive bubble technique. The films consisted of a two-dimensional array of tightly packed, polydisperse micro- and nanocrystals of mainly trigonal and hexagonal shape.  The size of the bubble was orders of magnitudes larger than the size of the crystals. Highest contact angle

values were obtained for silver concentrations pAg = 4 – 7, namely 70° – 73° for the advancing liquid meniscus, 20° – 24° for the receding liquid meniscus and 45° – 50° for the intermediate advancing angle. Lowest values were obtained for pAg = 1 with 16° for the advancing, 9° for the receding, and 30° for the intermediate advancing angle and pAg = 13 with 16° for the advancing, 9° for the receding, and 9° for the intermediate advancing angle. The maximum value was realized close to the PZC with pAg = 5.4 ± 0.2. This can be expected if one assumes that the spreading of water will be enhanced when the surface

charge tends to orient the water molecules into a position that is energetically more favorable than the orientation of the bulk at the charged interface would be (Bijsterbosch and Lyklema, 1978).

**B6 Dissolution of AgI particles in the CLINCH and IMCA/ZINC chambers**

An approximate dissolution rate of particles can be calculated by the Noyes-Whitney equation (Dokoumetzidis and Macheras, 2006):

$$\frac{dM}{dt} = -\left(\frac{D}{h}\right) A (C_s - C_t) \tag{1}$$




where $dM/dt$ describes the mass loss per time, $D$ is the diffusion coefficient, $A$ is the surface area of the particle(s), $C_s$ and $C_t$ are the saturation concentration and the concentration at time $t$, respectively, and $h$ is a thin static liquid layer at the solid surface under steady state conditions.

Diffusion coefficients of ions in aqueous solutions increase with increasing temperature. For Ag⁺ the diffusion coefficient is

$D = 8.5 \cdot 10^{-6}$ cm²/s at 273 K and $16.6 \cdot 10^{-6}$ cm²/s at 298 K. The diffusion coefficient is slightly larger for I⁻ with $10.3 \cdot 10^{-6}$ cm²/s at 273 K and $20.0 \cdot 10^{-6}$ cm²/s at 298 K (Li and Gregory, 1974). For the liquid layer $h$, a value equal to the particle radius is often assumed (Sheng et al., 2008). Silver iodide is a sparingly soluble salt with a solubility product of around $10^{-16}$ at 298 K (Hiemstra, 2012; Lyklema, 1966). The solubility decreases with decreasing temperature and reaches a value $< 10^{-17}$ at 273 K (Lyklema, 1966). For an estimated saturation concentration of AgI in water of $C_s = 7.4 \cdot 10^{-10}$ g/cm³ at T < 273 K and assuming $C_t = 0$, we obtain $dM/dt = 9 \cdot 10^{-19}$ g/s for a 200 nm particle and dM/dt = $9 \cdot 10^{-20}$ g/s for a 20 nm particle. To reach saturation concentration in a 80 µm droplet at 273 K, $2 \cdot 10^{-16}$ g AgI need to dissolve. With 2 s or 4 s residence time in the chamber, it can be assumed that the particle is constantly dissolving while in the chamber. The situation is different in the IMCA/ZINC chamber. Activation of the AgI particles to cloud droplets occurs at T > 273 K and the droplets grow only to 20 µm. Therefore, only $3 \cdot 10^{-18}$ g AgI need to dissolve to reach saturation concentration in the droplet at 273 K. At 298 K, dissolution occurs at a higher rate of $dM/dt = 6 \cdot 10^{-18}$ g/s for a 200 nm particle and at $dM/dt = 6 \cdot 10^{-19}$ g/s for a 20 nm particle. Residence time in the IMCA part is approximately 15 s. Therefore, there seems to be enough time to reach saturation within the IMCA section. The droplets might even become supersaturated with respect to AgI within the ZINC section, because dissolution occurred at T > 273 K and the saturation concentration is higher at this temperature than at the temperature in the ice nucleation section.

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




**Table 1.** Immersion freezing studies with AgI or related salts: $D_p$: particle diameter; $D_D$: droplet diameter; $C_p$: particle concentration; $C_D$: droplet concentration, $T_f$: freezing temperature. References: Ed62: Edwards et al., 1962; EE68: Edwards and Evans, 1968; EE62: Edwards and Evans, 1962; Zo08: Zobrist et al., 2008; Ag82: Aguerd et al., 1982; Ho61: Hoffer; SG72: Sax and Goldsmith, 1972; DM95: DeMott, 1995; 1961; VB84: Vonnegut and Baldwin, 1984; Pa86: Palanisamy et al., 1986b; Da75: Davis et al., 1975.

| IN | Preparation | Setup | $D_D$ & prep | $C_p$ | $D_p$ | Activation | $T_f$ | Refs |
|---|---|---|---|---|---|---|---|---|
| AgI | Precipitation from AgI saturated solution on seeds, deposited on cover slips | Cold stage; droplets on hydrophobic glass cover slips | $30 - 60\,\mu m$, av. $50\,\mu m$ sprayed from atomizer on cover slips | 10/drop | 170 nm | 50 % | 255 K | Ed62 |
| | | | | 100/drop | 170 nm | 50 % | 260 K | |
| | | | | 2/drop | 750 nm | 50 % | 258 K | |
| | | | | 10/drop | 750 nm | 50 % | 260 K | |
| | | | | 100/drop | 750 nm | 50 % | 263 K | |
| | | | | 2/drop | $3.5\,\mu m$ | 50 % | 263 K | |
| | | | | 20/drop | $3.5\,\mu m$ | 50 % | 266 K | |
| AgI | Precipitation from AgI saturated solution on seeds, deposited on cover slips | Cold stage; droplets on hydrophobic glass cover slips | $\sim 100\,\mu m$ sprayed from atomizer on cover slips | 1000/drop | $4\,\mu m$ | 50 % | 268 K | EE68 |
| | | | | 100/drop | $4\,\mu m$ | 50 % | 267.5 K | |
| | | | | 10/drop | $4\,\mu m$ | 50 % | 267K | |
| | | | | 2/drop | $4\,\mu m$ | 50 % | 266 K | |
| AgI | Precipitation from AgI saturated solution on seeds, deposited on cover slips | Cold stage; droplets on hydrophobic glass cover slips, sol A: excess $Ag^+$ | $30 - 60\,\mu m$, av. $50\,\mu m$ sprayed from atomizer on cover slips (~40) covered with paraffin oil | 200/drop | | | | | EE62 |
| | | | | $10^{-2}$ M AgNO$_3$ | 750 nm | 50 % | 257 K | |
| | | | | $10^{-3}$ M AgNO$_3$ | 750 nm | 50 % | 259 K | |
| | | | | $10^{-4}$ M AgNO$_3$ | 750 nm | 50 % | 262 K | |
| | | | | $10^{-5}$ M AgNO$_3$ | 750 nm | 50 % | 263 K | |
| | | | | $10^{-6}$ M AgNO$_3$ | 750 nm | 50 % | 259 K | |
| | | | | H$_2$O | 750 nm | 50 % | 255 K | |
| | | | | $10^{-3}$ M KI | 750 nm | 50 % | 255 K | |
| | | | | $10^{-4}$ M KI | 750 nm | 50 % | 255 K | |
| | | | | $10^{-5}$ M KI | 750 nm | 50 % | 255 K | |
| AgI | Precipitation from AgI saturated solution on seeds, deposited on cover slips | Cold stage; droplets on hydrophobic glass cover slips sol B: excess I$^-$ | $30 - 60\,\mu m$, av. $50\,\mu m$ sprayed from atomizer on cover slips (~40) | 200/drop | | | | | EE62 |
| | | | | $10^{-2}$ M AgNO$_3$ | 750 nm | 50 % | 252 K | |
| | | | | $10^{-4}$ M AgNO$_3$ | 750 nm | 50 % | 258 K | |
| | | | | $10^{-5}$ M AgNO$_3$ | 750 nm | 50 % | 263 K | |
| | | | | H$_2$O | 750 nm | 50 % | 264 K | |
| | | | | $10^{-6}$ M KI | 750 nm | 50 % | 263 K | |
| | | | | $10^{-5}$ M KI | 750 nm | 50 % | 257 K | |
| | | | | $10^{-4}$ M KI | 750 nm | 50 % | 256 K | |
| AgI, KNO$_3$ | In situ precipitation | Emulsions cooled in DSC | $3\,\mu m \pm 2\,\mu m$ in oil | 1/drop (assumed) | 260 nm | onset | 255 K | Zo08 |
| | | | | | 440 nm | $\sim 50$ % | 251.5 K | |
| AgI, KNO$_3$ | In situ precipitation | Emulsions cooled in DSC | $<3\,\mu m$ in oil | 1/drop (assumed) | 240 nm | onset | 255 K | Ag82 |
| | | | | | 240 nm | $\sim 50$ % | 251 K | |
| AgI, KNO$_3$ | In situ Precipitation | bulk cooled in DSC | 1 cm$^3$ | $10^{10}$/sample (assumed) | 200 nm | onset | 269.5 K | Ag82 |
| AgI, NaNO$_3$ | Precipitation in high viscosity silicone oil | Droplets embedded in oil, cooling at 1K/min | $100 - 120\,\mu m$ | Not known | Not known | 50 % | 257 K | Ho61 |





| | | | | | | | | |
|---|---|---|---|---|---|---|---|---|
| AgI | Heated wire with coated AgI | Chamber with falling droplets; 1 cm coagulation region; particles added at T > 273K | 40 – 160 µm; av. 100 µm stable stream | 100/drop, captured in coagulation region | 30 nm 30 nm | <1 % 50 % | 263 K 256 K | SG72 |
| AgI-AgCl | Burning acetone based solutions | Cloud chamber $C_D < 400$ cm$^{-3}$ | CCN activation of AgI particles (>50 %) at T > 268 K | 1/drop | 30 nm 30 nm 30 nm 30 nm 30 nm 70 nm 70 nm 70 nm 70 nm 70 nm | 0.05 % 0.5 % 0.8 % 2 % 5 % 0 % 0.01 % 0.1 % 0.2 % 1 % | 267 K 265 K 263 K 261 K 257 K 267 K 265 K 263 K 261 K 257 K | DM95 |
| AgI | Used as is | Suspension in U-shaped glass capillary tube with ca. 0.5 mm diameter | 0.01 g $H_2O$ | Many particles refreeze experiments at $T_{const}$ | 10 µm | 1 s 30 s 15 min 2 h | 264 K 266 K 268 K 270 K | VB84 |
| AgI | 99.95 % AR grade | Test tube filled with suspension, cooled with 0.1 K/min | 0.25 ml suspension | 0.25 mg per 0.25 ml suspension | ~ 55 µm | Repeated bulk sample freezing | 270.60 K | Pa86 |
| AgI-AgCl | 65:35 mol% AgI and AgCl molten together, then quenched to RT, powdered and sieved. | Test tube filled with suspension, cooled with 0.1 K/min | 0.25 ml suspension | 0.25 mg per 0.25 ml suspension | ~ 55 µm | Repeated bulk sample freezing | 271.97 K | Pa86 |
| AgI | – | Layer mounted on thermoelectric device | – | – | – | threshold | 269 K | Da75 |
| 3AgI· NH$_4$I· H$_2$O | From aqueous solution of AgI and NH$_4$I | Layer mounted on thermoelectric device | – | – | – | threshold | 272 K | Da75 |
| AgI, KNO$_3$ | Precipitated suspension, atomized and size selected | IMCA/ZINC chamber, activation section followed by nucleation section | 20 µm from AgI particles activated at T > 273K, $S_w > 20\%$ | 1/drop, 3 s residence time<br><br>1/drop, 10 s residence time | 200 nm 200 nm 200 nm 400 nm 20 nm 30 nm 40 nm 50 nm 200 nm 400 nm | 10 % 50 % 100 % 50 % 50 % 50 % 50 % 50 % 50 % 50 % | 265 K 262 K 255 K 263 K 236.5 K 259 K 264 K 264 K 264 K 264 K | This study |



**Table 2.** Contact freezing studies with AgI and related salts. $D_p$: particle diameter ; $D_D$: droplet diameter; $C_p$: particle concentration; $C_D$: droplet concentration, $T_f$: freezing temperature. References: GG68: Gokhale and Goold, 1968; GL71: Gokhale and Lewinter, 1971; SG72: Sax and Goldsmith, 1972; DM95: DeMott, 1995; La78: Langer et al., 1978; SF79: Schaller and Fukuta, 1979; Na16: Nagare et al., 2016.

| IN | Preparation | Setup | $D_D$ & prep | $C_p$ | $D_p$ | Activation | $T_f$ | Refs |
|---|---|---|---|---|---|---|---|---|
| Ag mixed with NaI | Burner smoke from AgI string generator | Droplets on hydrophobic plate, cooling at 1.3 K/min | 2.7 mm deposited | Smoke (introduced in chamber at 268 K) | 50 – 100 nm | 4 % 50 % | 268 K 263 K | GG68 |
| AgI | Crushed powder | Droplets on hydrophobic plate | 2.7 mm deposited | many particles (sprinkled on drops at 268 K) | 4 – 10 µm 5 – 20 µm 100 – 400 µm | 100 % 100 % 100 % | 268 K 268 K 268 K | GG68 |
| AgI, used as is | Purified grade | Droplets on hydrophobic plate | 2 mm deposited | many particles (sprinkled on drops at 268 K) | micron size | Freezing within 16 – 47 ms | 268 K | GL71 |
| AgI | Heated wire coated with AgI | Chamber with falling droplets; 1 cm coagulation region; particles added at $T_f$ | 40 – 160 µm; av. 100 µm stable stream | 100/drop, captured in coagulation region | 30 nm 30 nm 30 nm | 1 % 50 % 100 % | 263 K 258 K 256 K | SG72 |
| AgI | Heated wire with coated AgI | Chamber, particles added at target temperature | 1 mm on tip of fiber in chamber | $10^5$/drop < 250/drop | 30 nm 30 nm | 50 –100 % 50 –100 % | 268 K 263 K | SG72 |
| AgI-AgCl | Burning acetone based solutions | Isothermal cloud chamber (960 L) $t_{exp}$= 3 – 30 min $C_D$ = 4300 cm$^{-3}$ | 8 µm (1 – 16 µm) Continuously produced by nebulizing water | 1/drop (nominal) | 30 nm 30 nm 30 nm 30 nm 30 nm 70 nm 70 nm 70 nm 70 nm 70 nm | 0.04 % 6.6 % 12.2 % 24.3 % 34.4 % 0.2 % 39.8 % 54.6 % 45.8 % 65.0 % | 267 K 265 K 263 K 261 K 257 K 267 K 265 K 263 K 261 K 257 K | DM95 |
| AgI | Thermally (from molten AgI) | Cloud chamber, Residence time: 120 s (20 – 300 s) | av. 6 µm (1 – 10 µm) activated from NaCl aerosol, $C_D$ = 4·10$^4$ cm$^{-3}$ | 100 – 1000 cm$^{-3}$ air | 20 nm 40 nm 60 nm 80 nm 100 nm 120 nm | 1.2 % 15 % 28 % 41 % 46 % 47 % | 257–258 K 257–258 K 257-258 K 257-258 K 257-258 K 257-258 K | La78 |
| AgI, | Reagent grade heated on iron plate | Wedge-shaped ice thermal diffusion chamber | Small fog droplets activated from NH$_4$Cl particles | 1/drop | Average: 300 nm | not known | 267 K | SF79 |
| AgI, KNO$_3$ | Precipitated suspension atomized and size selected | CLINCH Continuous flow diffusion chamber | 80 µm injected in chamber | 1/drop 1/drop 2/drop 2/drop | 200 nm 200 nm 200 nm 200 nm | 50 % 100 % 50 % 100 % | 248 K 246 K 258 K 250 K | Na16 |





**Table 3.** Condensation freezing studies. $D_p$: particle diameter; $D_D$: droplet diameter; $C_p$: particle concentration; $C_D$: droplet concentration, $T_f$: freezing temperature. $S_w$: supersaturation with respect to water. References: SF79: Schaller and Fukuta, 1979; EE60: Edwards and Evans, 1960; Zi07: Zimmermann et al., 2007.

| IN | Preparation | Setup | $D_D$ & prep | $C_p$ | $D_p$ | Activation | $T_f$ | Refs |
|---|---|---|---|---|---|---|---|---|
| AgI, | Reagent grade heated on iron plate | Wedge-shaped ice thermal diffusion chamber | Small fog droplets activated from AgI at $S_w$= 12.5% | 1/drop | av: 300 nm | 1.3 % <br> 6.4 % | 268.5 K <br> 267 K | SF79 |
| AgI | Ag aerosol reacting with iodine vapour | Cloud chamber | Activation at $S_w$= 20 % | 1/drop | 12 – 15 nm <br> 18 – 20 nm | onset <br> onset | 264 – 265 K <br> 266 – 267 K | EE60 |
| AgI, p.a. | grinded | ESEM, increasing RH in steps of 0.13 hPa (5 – 10 min) | Activation at $S_w$ = 8 – 11% | 1/drop | 1 – 10 µm | threshold | 268 K | Zi07 |



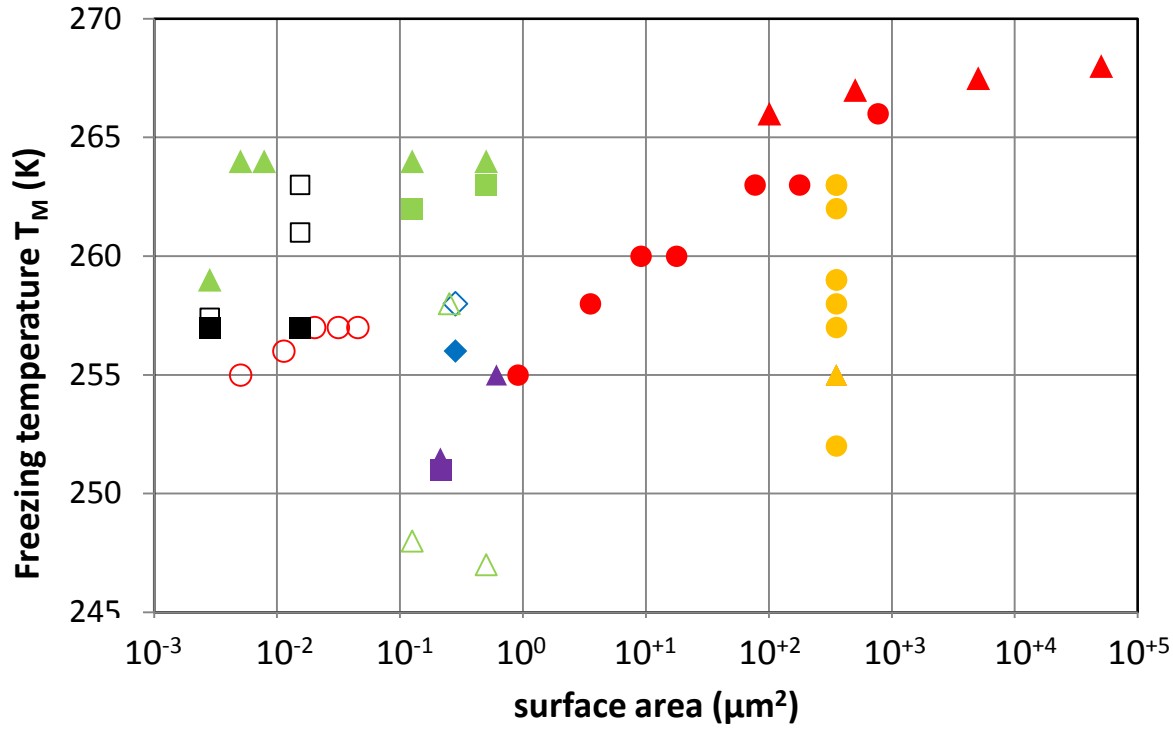

**Figure 1.** Median freezing temperatures $T_M$ as a function of particle surface area per droplet for contact freezing (open symbols) and immersion freezing (filled symbols). Filled red symbols: droplets on hydrophobic glass slips with previously deposited AgI particles prepared by precipitation in stoichiometric ratio (triangles: Edwards and Evans, 1968; circles: Edwards et al., 1962). Filled orange symbols: droplets on hydrophobic glass slips with previously deposited AgI particles prepared by precipitation in non-stoichiometric ratio (Edwards and Evans, 1962; circles: excess of $AgNO_3$, triangles: excess of KI). Filled purple symbols: emulsion of AgI containing water droplets prepared by in-situ precipitation in stoichiometric ratio (triangles: Zobrist et al., 2007; square: Aguerd et al., 1982). Filled green symbols: Freely falling water droplets with immersed AgI particles (squares: this study, 3 s residence time; triangles: this study, 10 s residence time). Filled black squares: cloud chamber filled with droplets from CCN activation of AgI-AgCl particles (DeMott, 1995). Open black squares: water droplets in cloud chamber with AgI-AgCl aerosol (DeMott, 1995). Open green symbols: water droplets freely falling through chamber with AgI aerosol (Nagare et al., 2016). Open red circles: water droplets in cloud chamber filled with AgI aerosol (Langer, 1978). Blue diamonds: freely falling droplets intercepting AgI aerosol, filled diamond: immersion mode, open diamond: contact mode (Sax and Goldsmith, 1972).




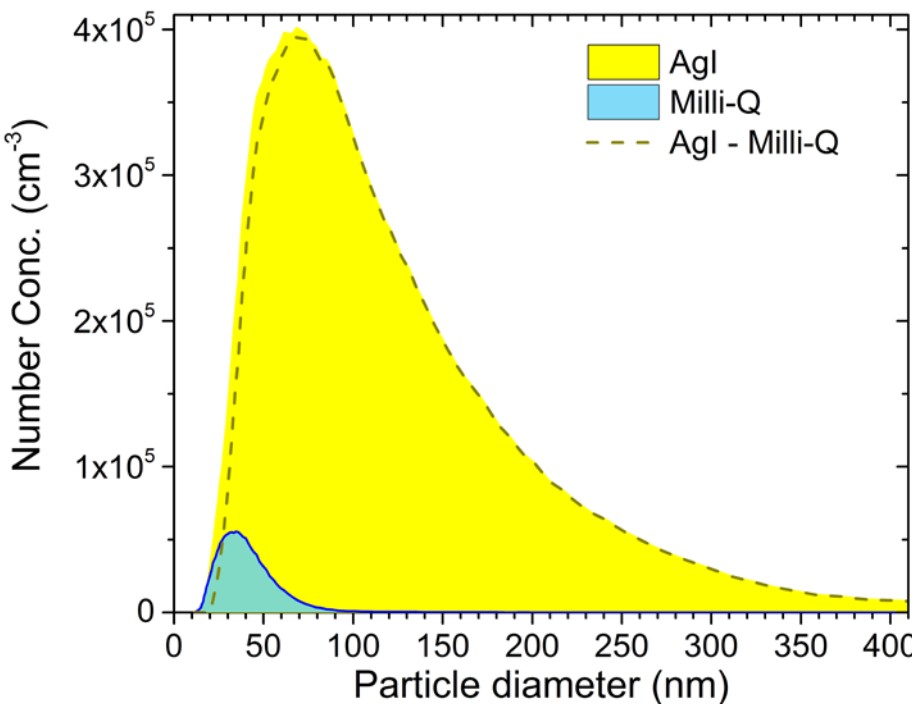

**Figure A1.** Size distribution of an AgI suspension in Milli-Q water (yellow area) measured with a TSI 3080 scanning mobility particle sizer (SMPS). Pure Milli-Q water measured with the same method and settings is shown as the blue area. The difference curve area(AgI) – area(Milli-Q) is given as dashed line.





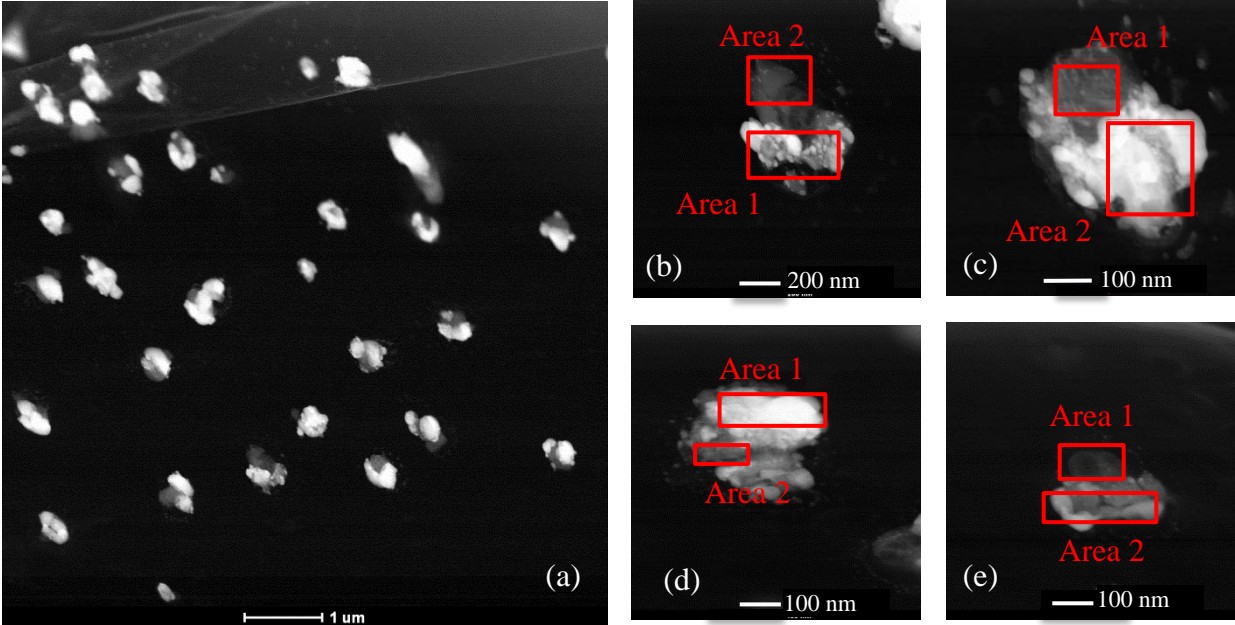

**Figure A2.** High angle annular dark field (*HAADF*) scanning transmission electron microscopy (*STEM*) of AgI particles formed by precipitation. Panel (a) shows an overview and panels (c – e) individual particles for which EDX (energy dispersive X-ray) spectroscopy has been performed. Panel (b): In area 1 silver and iodine was detected with small amounts of oxygen and potassium. In area 2 potassium and chloride were detected. Panel (c): In area 1 silver and iodine was detected, for area 2 the EDX spectrum showed a large peak for potassium and small peaks for chlorine and oxygen. Panel (d): Area 1 contained mainly silver and iodine, area 2 silver and iodine and possibly small amounts of potassium. Panel (e): The EDX spectrum of area 1 showed a small peak for potassium and small amounts of silver and iodine, area 2 silver and iodine.





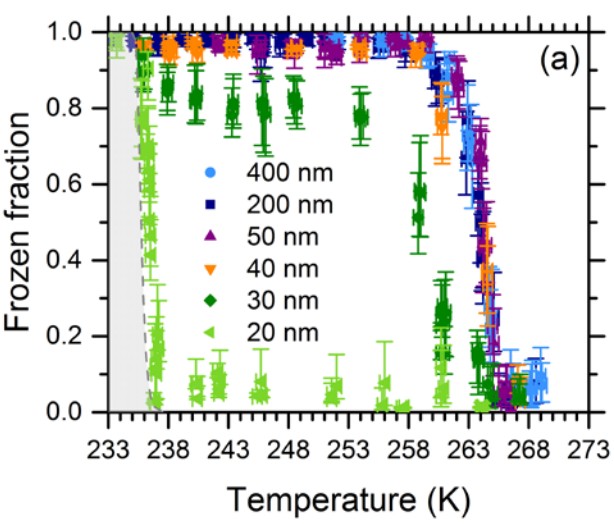
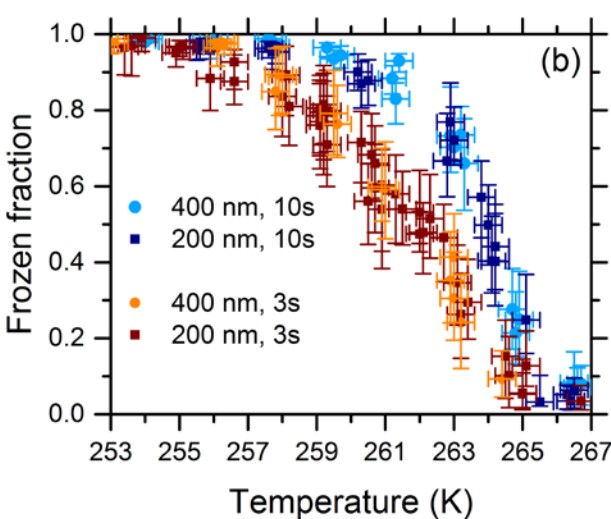

**Figure A3.** Immersion freezing experiments performed with the IMCA/ZINC setup. Panel (a): particle size dependence of frozen fraction. Panel (b): time dependence. Note, that the scales of the x-axes in panels (a) and (b) are different. Error bars represent the uncertainty in the frozen fraction due to the classification (liquid or ice) uncertainty of the IODE detector (Lüönd et al., 2010).