# Peer review of "Ice nucleation efficiency of AgI: review and new insights"

_Atmospheric Chemistry and Physics, 2016_

## Referee Comment (RC1) · Anonymous Referee #1 · 5 Apr 2016

I have reviewed the article "Ice nucleation efficiency of AgI: review and new insights" by Marcolli et al. This article covers literature dating back to the 1960s on ice nucleation on AgI. The article focuses on the molecular level processes that may cause AgI to be an efficient ice nucleus. The article is timely, in that there is increasing interest in the processes that lead to efficient ice nucleation, and it would be of great interest to the readers of Atmos. Chem. Phys. I have a number of minor suggestions for authors to consider, which are listed below:

p2 line 7: It is interesting that condensation freezing is considered separately from immersion. To what extent does condensation freezing have a different efficiency than immersion? Can it also take place below water saturation, like deposition nucleation?

pg 2 line 32: Is having a defect enough even if none of these "requirements" are met?

[Figure]

pg 2 line 31: It's odd to refer to these as requirements. Are they requirements, or simply properties that promote ice nucleation?

pg 3 line 28: You should specify if the symbols in Fig 1 are filled or open.

pg 4 line 6: What is meant by "adsorb iodide ions on the surface"? Does it mean that iodide in the solution is surface active or something else?

Fig. 1: Indicate that the orange circles and triangles overlap.

pg 11 line 25: If AgI particles stay on the surface of the droplets during immersion freezing, how is an "inside-out" contact freezing mechanism being separated from immersion freezing?

pg 11 line 1: A reference is needed here. Water molecules at the interface participate in fewer H-bonds and therefore may have greater mobility. In polymers for example, there is greater mobility at the air-polymer interface than in the bulk of the polymer.

pg 12 line 13: Why should the amount of water adsorbed on a surface at a given RH matter for immersion freezing, where the nucleation presumably takes place at a location on the heterogeneous nucleus inside the water droplet?

pg 12 line 15: I do not understand this result for condensation freezing. Again, the surface is immersed in a solution in this mechanism, so it is hard to believe that surface adsorbed water due to RH has an effect. Activation may mean that the solution is most dilute, allowing freezing to occur.

pg 12 line 31: Why is the observed dissolution greater for the contact freezing experiment (with the same particle diameter)? Also, is this mobility diameter?

pg 13 line 17: As long as the surface is dissolved, the ice nucleation activity should be reduced according to the arguments in this section. Why should the amount of dissolution matter? Do you suspect that at 0.04% dissolution the surface is not fully dissolved? Are these arguments consistent with the comparison of the two particle

sizes?

pg 14 line 8: Why does the addition of AgCl increase the ice nucleation activity? (This is stated in the conclusions, but it should be stated here as well.)

Fig. A2: SAED diffractograms would be interesting to report to see the amount of different polymorphs in your sample. Were these taken?

Wording/Grammar: pg 3 line 9: "Such an analysis should also allow us to identify factors"

pg 6 line 27: "They noted that a supersaturation of 20%...."

pg 8 line 29 & 30: For Miller indices, it is more typical to write 001 with a bar over the 1 rather than 00-1.

pg 10 line 21: What is meant by "in sum"?

pg 15 line 14: change "with different shares" to "in different proportions"
* * *

---

## Referee Comment (RC2) · Anonymous Referee #2 · 19 Apr 2016

General comment:

The paper summarizes results about the ice nucleating efficiency of silver iodide in different modes obtained by various experimental techniques. Studies from several decades are summarized and explanations about the reasons of the efficient ice nucleating ability of silver iodide are described in molecular levels. Such a review gives important scientific insights and findings and is of high scientific relevance.

However, representing such a review is not a trivial task. The descriptions of the utilised techniques are not given in a way that the reader could easily follow (which is, of course, a challenge). I see some weaknesses in defining the different ice nucleating processes in their differences. The discussion about the stochastic nature of freezing processes and the neglecting of time dependence should be more detailed. Furthermore, I find

it critical that the paper mixes the review of previous findings with new experimental results which have not been published elsewhere. This should be treated separately in the paper, such as starting with an experimental part with new results, and then continue with the review part.

To my opinion, major revisions regarding the presentation quality are requested before publishing the manuscript in APC.

Specific comments

1. The paper should be rearranged so that previous findings are clearly separated from new experimental results which have not been published elsewhere, such as starting with an experimental part with new results, and then continue with the review part.

2. Clear definitions of the treated freezing modes should be given in a section in the Review part.

3. The descriptions of the utilised techniques should be completely reworked. Maybe it would be better not to describe too many details in the text which are listed in the tables.

4. In the text the use of the terms ice nuclei and IN, ice nucleating particles and INP is not consistent. Please correct this.

5. Temperatures are sometimes given in °C, sometimes in K. Please change this consistently.

6. Abstract: Including full references with all details in a paper is not usual; the same is the case for including references at all in the Abstract.

7. Abstract, page 2, line 10: Deposition freezing: What about experimental results from deposition freezing with AgI particles?

8. Abstract, and Introduction, page 3: The remark that "this paper is one of three papers that present and analyse contact freezing experiments with AgI" is somehow

confusing. Unpublished results should be clearly presented in the paper as suggested, previous results should be treated equally all together.

9. Introduction, page 2, line 8: There are recent studies showing that the freezing temperature in the contact mode is dependent on the particle size, e.g., Hoffmann et al., 2013, Faraday Discuss, 165. Therefore, only for large particle sizes contact freezing temperatures are higher than for immersion freezing.

10. Introduction, page 2, line 15: I would suggest to start a new paragraph here.

11. Page 3, line 21, and many other places: In the text, there is often written something like "range from x – y" or "diameters of x – y". Please avoid using "– " in the text and write, e.g. "range from x to y".

12. Page 8, line 10: The sense of the last sentence of this paragraph is not clear.

13. Page 8, line 14: What is meant by "even stronger dependence"?

14. Page 10, line 27: Please reformulate the sentence " simulations . . . were able to simulate . . .".

15. Page 11, Section 3.4, and other places: Please replace "totally" by "completely" or "entirely".

16. Figure 1: At least some short explanations of the symbols included directly in the figure would help the reader.

---

## Author Comment (AC1) · 2 Jun 2016

**Response to anonymous Referee #1**

*We thank the reviewer for the positive review and the careful reading of the manuscript. The comments are addressed below in italic.*

I have reviewed the article "Ice nucleation efficiency of AgI: review and new insights" by Marcolli et al. This article covers literature dating back to the 1960s on ice nucleation on AgI. The article focuses on the molecular level processes that may cause AgI to be an efficient ice nucleus. The article is timely, in that there is increasing interest in the processes that lead to efficient ice nucleation, and it would be of great interest to the readers of Atmos. Chem. Phys. I have a number of minor suggestions for authors to consider, which are listed below:

p2 line 7: It is interesting that condensation freezing is considered separately from immersion. To what extent does condensation freezing have a different efficiency than immersion? Can it also take place below water saturation, like deposition nucleation?

*Condensation freezing occurs concurrent with the activation of a particle to a cloud droplet and its efficiency might be different from the one of immersion freezing. A definition of condensation freezing is also given in Vali et al. (2015). Although it is not clear whether immersion and condensation freezing have different efficiencies, these two modes are usually discriminated. We added a section in the revised manuscript that describes the different modes of heterogeneous ice nucleation.*

pg 2 line 32: Is having a defect enough even if none of these "requirements" are met?

*It is hard to imagine a defect that does not influence surface charge, polarizability, hydrogen bonding or van der Waals interactions. Therefore, when there is a defect, at least some of these properties change.*

pg 2 line 31: It's odd to refer to these as requirements. Are they requirements, or simply properties that promote ice nucleation?

*We agree that "requirements" might be a too strong expression in this context and replace it by "properties".*

pg 3 line 28: You should specify if the symbols in Fig 1 are filled or open.

*They are filled. This is now stated in the text of the revised manuscript.*

pg 4 line 6: What is meant by "adsorb iodide ions on the surface"? Does it mean that iodide in the solution is surface active or something else?

*This is explained in Appendix B3: "Because of the better solubility of $Ag^+$ compared to $I^-$, a neutral AgI crystal will be negatively charged in water or aqueous solutions." Surface activity is not meant. To make this clearer, we replace "adsorb" by "enrich":*

Fig. 1: Indicate that the orange circles and triangles overlap.

*This is done in the revised manuscript.*

pg 11 line 25: If AgI particles stay on the surface of the droplets during immersion freezing, how is an "inside-out" contact freezing mechanism being separated from immersion freezing?

*In continuous flow diffusion chambers, particles freely float in a droplet and can take a position on the droplet surface or immersed in the droplet depending on the balance of interfacial and surface forces. Therefore, an experiment that is intended to study immersion freezing can turn into a contact freezing experiment. This circumstance has not gained much attention yet. To determine whether immersion or contact freezing inside-out is at work, the position of the particle in or on the droplet must be derived based on the wetting behavior of the particle. This is discussed in more detail in the companion paper by Nagare et al. (2016).*

pg 11 line 1: A reference is needed here. Water molecules at the interface participate in fewer H-bonds and therefore may have greater mobility. In polymers for example, there is greater mobility at the air-polymer interface than in the bulk of the polymer.

*The reference is Taylor and Hale (1993).*

pg 12 line 13: Why should the amount of water adsorbed on a surface at a given RH matter for immersion freezing, where the nucleation presumably takes place at a location on the heterogeneous nucleus inside the water droplet?

*If ice nucleation happened on the surface that is totally immersed in water, RH should not matter. However, the studies compiled in Fig. 1 show that freezing temperatures*

*are higher for setups where the particle can access the water surface compared with experimental arrangements where the droplet is immersed in or covered with oil. This indicates that freezing is not occurring on the surface that is totally immersed in water but either on the contact line to air or in water patches on the surface exposed to air. Water patches on the surface are indeed influenced by RH.*

pg 12 line 15: I do not understand this result for condensation freezing. Again, the surface is immersed in a solution in this mechanism, so it is hard to believe that surface adsorbed water due to RH has an effect. Activation may mean that the solution is most dilute, allowing freezing to occur.

*The argumentation is based on the assumption that ice nucleation occurs on water patches on the surface exposed to air.*

pg 12 line 31: Why is the observed dissolution greater for the contact freezing experiment (with the same particle diameter)? Also, is this mobility diameter?

*The particle diameter is the same but the droplet diameter is different. Droplets in IMCA/ZINC are 18 – 20 µm in diameter, the ones in CLINCH are 80 µm in diameter. Therefore, dissolution is greater for the contact freezing experiment in CLINCH than for the immersion freezing experiment in IMCA/ZINC. The particles are size selected according to mobility diameter.*

pg 13 line 17: As long as the surface is dissolved, the ice nucleation activity should be reduced according to the arguments in this section. Why should the amount of dissolution matter? Do you suspect that at 0.04% dissolution the surface is not fully dissolved? Are these arguments consistent with the comparison of the two particle sizes?

*The amount of dissolution should not matter but it should matter whether a particle is in the process of dissolving, because the surface should be different for a dissolving particle that is not in equilibrium with the surrounding solution compared with a particle that has reached or almost reached equilibrium with the solution. Equilibrium is reached or almost reached in the case of the immersion freezing experiments because activation to a water droplet occurs already in the IMCA section at warmer temperature. In the CLINCH experiment, a particle that collided with a droplet is continuously dissolving while it passes through the chamber. This is explained in Appendix B6.*

pg 14 line 8: Why does the addition of AgCl increase the ice nucleation activity? (This is stated in the conclusions, but it should be stated here as well.)

*In the presence of AgCl, AgI-AgCl solid solutions are formed, which have a better lattice match with ice than pure AgI. This is discussed in Sect. 3.3. We now mention this in the revised manuscript and refer to Sect. 3.3.*

Fig. A2: SAED diffractograms would be interesting to report to see the amount of different polymorphs in your sample. Were these taken?

*No, we just measured XRD diffractograms of the bulk sample that revealed a mixture of the $\beta$- and the $\gamma$-phase.*

Wording/Grammar: pg 3 line 9: "Such an analysis should also allow us to identify factors" *Done*

pg 6 line 27: "They noted that a supersaturation of 20%...." *Done*

pg 8 line 29 & 30: For Miller indices, it is more typical to write 001 with a bar over the 1 rather than 00-1. *Done.*

pg 10 line 21: What is meant by "in sum"?

*"in sum" is misplaced. We rearranged the text and put it in front of "to low configurational entropy"*

pg 15 line 14: change "with different shares" to "in different proportions" *Done*

*Reference:*

*Vali, G., DeMott, P. J., Möhler, O., and Whale, T. F.: Technical Note: A proposal for ice nucleation terminology, Atmos. Chem. Phys., 15, 10263–10270, doi:10.5194/acp-15-10263-2015, 2015.*

---

## Author Comment (AC2) · 2 Jun 2016

*We thank the reviewer for the careful reading of the manuscript and the suggestions for improvement. The comments are addressed below in italic.*

General comment:

The paper summarizes results about the ice nucleating efficiency of silver iodide in different modes obtained by various experimental techniques. Studies from several decades are summarized and explanations about the reasons of the efficient ice nucleating ability of silver iodide are described in molecular levels. Such a review gives important scientific insights and findings and is of high scientific relevance.

However, representing such a review is not a trivial task. The descriptions of the utilised techniques are not given in a way that the reader could easily follow (which is, of course, a challenge). I see some weaknesses in defining the different ice nucleating processes in their differences. The discussion about the stochastic nature of freezing processes and the neglecting of time dependence should be more detailed. Furthermore, I find it critical that the paper mixes the review of previous findings with new experimental results which have not been published elsewhere. This should be treated separately in the paper, such as starting with an experimental part with new results, and then continue with the review part.

To my opinion, major revisions regarding the presentation quality are requested before publishing the manuscript in APC.

*We improved the structure of the manuscript as suggested by the reviewer and added a paragraph at the end of the introduction to outline the structure to the reader.*

Specific comments

1. The paper should be rearranged so that previous findings are clearly separated from new experimental results which have not been published elsewhere, such as starting with an experimental part with new results, and then continue with the review part.

*We moved the Appendix A1 to the main text as an experimental part and rearranged the main text as suggested.*

2. Clear definitions of the treated freezing modes should be given in a section in the Review part.

*We added Sect. 3.1 (Modes of heterogeneous ice nucleation) to the revised manuscript.*

3. The descriptions of the utilised techniques should be completely reworked. Maybe it would be better not to describe too many details in the text which are listed in the tables.

*We carefully went through the text and improved it.*

4. In the text the use of the terms ice nuclei and IN, ice nucleating particles and INP is not consistent. Please correct this.

*In the revised manuscript, we avoid using an abbreviation.*

5. Temperatures are sometimes given in °C, sometimes in K. Please change this consistently.

*We converted all temperatures to Kelvin.*

6. Abstract: Including full references with all details in a paper is not usual; the same is the case for including references at all in the Abstract.

*We delete the references from the abstract in the revised version.*

7. Abstract, page 2, line 10: Deposition freezing: What about experimental results from deposition freezing with AgI particles?

*Our main focus was to compare ice nucleation efficiency at water saturation from studies performed in contact, immersion, and condensation mode. We agree that a review on ice nucleation by AgI should also treat deposition nucleation. Therefore, we added a Table 4 that summarizes studies performed in deposition mode and discuss them in more detail in the text.*

8. Abstract, and Introduction, page 3: The remark that "this paper is one of three papers that present and analyse contact freezing experiments with AgI" is somehow confusing. Unpublished results should be clearly presented in the paper as suggested, previous results should be treated equally all together.

*We delete the references from the abstract in the revised version.*

9.  Introduction, page 2, line 8: There are recent studies showing that the freezing temperature in the contact mode is dependent on the particle size, e.g., Hoffmann et al., 2013, Faraday Discuss, 165. Therefore, only for large particle sizes contact freezing temperatures are higher than for immersion freezing.

*Hoffmann et al. observed a surface area dependence for contact freezing. Such a dependence is also expected for immersion freezing. Hence, freezing efficiencies in immersion and contact mode should show the same size dependence.*

10.  Introduction, page 2, line 15: I would suggest to start a new paragraph here.

*This is done.*

11.  Page 3, line 21, and many other places: In the text, there is often written something like "range from x – y" or "diameters of x – y". Please avoid using "– " in the text and write, e.g. "range from x to y".

*We changed the text as suggested.*

12.  Page 8, line 10: The sense of the last sentence of this paragraph is not clear.

*To make the meaning of the last sentence clearer, we insert the following sentence: "Thus, the difference in temperature between a frozen fraction of 0.1 and a fully frozen sample is 8 K."*

13.  Page 8, line 14: What is meant by "even stronger dependence"?

*We add to the sentence: "An even stronger dependence of freezing temperature…"*

14.  Page 10, line 27: Please reformulate the sentence " simulations . . . were able to simulate . . .".

*We reformulated this sentence: "In agreement with this, Fraux and Doye (2014) and Zielke et al. (2015) were able to simulate ice growth …"*

15.  Page 11, Section 3.4, and other places: Please replace "totally" by "completely" or "entirely".

*We replaced "totally" by "completely".*

16.  Figure 1: At least some short explanations of the symbols included directly in the figure would help the reader.

*We inserted a legend to the right of the Figure in the revised manuscript.*

---

## Author Response (AR2)

Dear Daniel

Thank you for the careful reading of our manuscript.

Below are our responses written in italic.

Best wishes

Claudia

Page 2, line 6: If you cite Zobrist et al., 2007, wouldn't it be fair to also cite these studies on this topic: Cantrell, W.; Robinson, C. Geophys. Res. Lett. 2006, 33, L07802, Knopf and Forrester, J. Phys. Chem. A 2011, 115, 5579–5591 ?

*We added these references.*

Page 3, line 24 following: "Therefore,…". When reading the text, I had the impression you talk about immersion freezing (IMCA/ZINC) but then you argue with contact ice nucleation for particles < 40 nm. Maybe I am just missing something. The section title mentions IMCA/ZINC which are the immersion freezing experiments and Figure 1 is discussed.

*This section is indeed about immersion freezing. We referred to Langer et al. (1978) as an example of 20 nm particles that were shown to be ice nucleating. Because the freezing mode is not important for this general statement, we remove this information to avoid confusion.*

Page 4, line 27 & Page 9, section 4: Figure 2: The median freezing temperatures are given. I believe it would be very beneficial to also plot the uncertainties such as 10 and 90 percentiles (or 25, 75, or 20, 80 percentiles). I would not be surprised that for some data sets, the trend is within the scatter/uncertainty of the freezing data. I am aware, that it may be difficult to derive for all data the percentiles (and this depends on the number of data points as well). In this case, maybe use experimental uncertainty (temperature error, etc.).

*We agree that error bars would be beneficial. However, the information provided in the different studies does not allow to derive any percentiles. Error bars would be arbitrary and therefore not helpful to judge the consistency between measurements.*

Page 9, line 16: I assume you meant Fig. 2?

*Yes, thank you for pointing this out.*

Page 20, B3: I believe Appendix B3 is not referred to in the main text.

*Thank you for pointing this out. We added the reference on page 12, line 10.*

Technical correction:
Page 4, line 6: "…supercooled liquid particle…"

*We added "supercooled"*